# Tidewater cycle drives alpine glacial sediment plume geochemistry

K. O. Forsch ®[1,2] ✉, A. Ruacho[3] & S. M. Aarons ®[1]

Sediments transported by glacial meltwaters are important sources of trace-metal micronutrients for coastal microbial communities, linking cryospheric processes with ocean fertilization and biogeochemical cycles. Tidewater glacier advance-retreat cycles drive sediment fluxes and influence fjord geochemistry. Here, we used a chemical extraction method to determine the iron and manganese fertilization potential of suspended sediment-plume and iceberg-laden particulate matter from two adjacent, yet geomorphologically distinct, fjords in Southcentral Alaska. We found that the glacier retreat status underpinned the fraction of labile trace-metals within fjord surface plumes, with distinctly lower lability of metals associated with recent and rapid retreat coincident with enhanced erosion and chemical weathering. Particle size did not affect chemical lability, resulting in a well-mixed particle assemblage transported to the coastal ocean. With global tidewater glacier retreat, these results provide implications for future ecosystem fertilization via cryospheric processes and the interpretation of fjord sediment archives.

The temperate coastline of Southcentral Alaska is a landscape dominated by mountains and fjords, which encircle the northern Gulf of Alaska and include land-terminating (also lake-terminating) and tidewater glaciers (marine-terminating). Tidewater fjords are unique ecosystems that occur in high-latitude coastal environments throughout Alaska, Antarctica, the Canadian Arctic, Russian Arctic, Greenland, Chile, Iceland, and Svalbard[1]. Most tidewater glaciers around the world are retreating due to global climate change and are predicted to continue to retreat[2], contributing to global sea level rise[3] and disturbance of marine ecosystems[4]. The Kenai fjords, located on the Kenai Peninsula in Southcentral Alaska, were carved by outlet glaciers flowing from the 2080 km² Harding Icefield[5]. Today, active tidewater glaciers exist in only three fjords, whereas the remaining glaciers are land-terminating, either connected or disconnected from the icefield[6]. A large emphasis has been placed on understanding how increased meltwater inputs and glacier degradation will impact primary production in coastal marine systems[7]. Increased meltwater discharge has the potential to adversely affect coastal marine microbial communities, whereby macronutrient (nitrate, $NO_3^-$ and phosphate, $PO_4^{3-}$) replenishment via offshore subsurface water masses decreases, and greater fluxes of suspended sediments attenuate photosynthetically

active light[8,9]. However, the delivery of dissolved trace-metal micronutrients (iron (Fe), manganese (Mn)) to coastal and offshore marine systems from glaciated catchments has garnered large efforts to quantify the concentration of trace metals in glacial meltwaters and fluxes to the ocean[10]. This motivation is due to the proximity of the polar cryosphere to micronutrient-poor shelf ecosystems (e.g., Gulf of Alaska, Southern Ocean)[11,12].

All glaciers are sensitive to ongoing climate change, but tidewater glaciers may be influenced by both climate forcing and fjord geomorphology, resulting in a unique pattern of advance and retreat. This natural cycle exists in glaciomarine systems, whereby tidewater glaciers periodically advance, reach a stable morphology, and rapidly retreat to a new stable position. This is known as the 'tidewater-glacier cycle' (TGC, hereafter) and has been empirically modeled, linking ice flow, meltwater fluxes, and sediment transport[13]. The centuries-long behavior of tidewater glaciers consists of recurring periods of advance followed by rapid glacial retreat with intermittent periods of stability. The influence of the TGC stage upon the chemistry of glacial sediment plumes has not yet been explored in detail. Due to the timescales required to observe changes in sediment dynamics related to this cycle (centuries to millennia), it is impossible to measure the geochemical

[1]Geosciences Research Division, Scripps Institution of Oceanography, University of California, San Diego, La Jolla, CA, USA. [2]Department of Earth Sciences, University of Southern California, Los Angeles, CA, USA. [3]School of Oceanography, University of Washington, Seattle, WA, USA. ✉e-mail: forsch@usc.edu

impact that the processes of the TGC have on a single fjord from sampling the water column. Tidewater glaciers are important vectors for material exchange between the continental crust and the ocean, and rival sediment fluxes associated with riverine inputs[14]. Subglacial buoyant plumes and flexture of the glacier terminus ('tidal pumping') enhance the delivery of sediments and bioessential elements to the fjord surface, including subsurface $NO_3^-$, $PO_4^{3-}$[15], and meltwater-associated dissolved silica (dSi)[16] and micronutrients[17]. Iceberg calving increases the distal impact of tidewater glaciers through transporting sediments further down-fjord[18,19]. These processes (e.g., subglacial plumes and iceberg calving) would cease when tidewater glaciers eventually retreat onto land, highlighting the temporal and spatial sensitivity of the exchange between land and the ocean in tidewater glacial environments.

Dissolved trace-metal micronutrient delivery to coastal marine systems from glaciated catchments is well studied[20–31], yet intense gradients in oxygenation, salinity, and turbidity act to reduce metal concentrations in the water column through the formation of authigenic particulate matter and aggregation[32]. These processes together are known as 'boundary scavenging' and the precipitated or co-precipitated phases are known to be chemically labile[24,28,30,33–35] and bioavailable[35–37]. Therefore, particulate matter derived from bedrock erosion, chemical weathering, and subsequent precipitation of authigenic minerals dominates micronutrient speciation and bioavailability in these locations. In most large glaciated regions, natural variability exists in bedrock composition, glacial geomorphology, and discharge rates. How the stage of the TGC affects the geochemistry of particulate matter transported to the coastal ocean remains untested. For example, a recent study of Greenland outlet glaciers estimated catchments of tidewater glaciers are on average 20% larger than those of land-terminating glaciers, with ice velocities twice as high as land-terminating glaciers, implying greater sediment production by erosion for tidewater glaciers and transport generated through surface runoff[14]. In contrast, land-terminating glaciers are typically characterized by more proglacial deposition processes (i.e., streams and rivers), further limiting their potential for sediment transport to the coastal ocean.

Alaska tidewater glaciers are losing mass at a much slower rate than other alpine glaciers[38]. Despite stability in the retracted state, most tidewater glaciers are still experiencing net mass loss, suggesting that the warming climate has suppressed the onset of glacial advancement[39]. Mass-balance studies at other tidewater glaciers throughout Alaska indicate that summer temperatures are driving negative mass balances through ablation processes such as large-scale areal retreat and elevation change[6,38]. The most notable physical changes in glaciers as they relate to fjord biogeochemistry are the transition of a tidewater glacier to land-terminating, which will reduce or completely shut off the glacial impacts upon fjord circulation through buoyant meltwater from the subglacial environment entering the fjord in the subsurface (i.e., at the grounding line)[4,15,40]. In these two archetypal configurations, the residence time of weathering products in subglacial hydrologic environments and subsequent transport to the ocean are hypothesized to dictate the transformation of labile particulate trace metals in fjord surface waters[41]. In contrast to tidewater glaciers, land-terminating glaciers are characterized by proglacial processes, and products of subglacial weathering are transported long distances through fluvial outwash plains, providing ample opportunity for chemical transformations such as changes in oxidation state or speciation. A tidewater glacier instead has englacial channels and short transport times between the altitude and position of maximum erosion to the subsurface marine environment, providing a more direct source-to-sink transport process.

To better understand the role of recent retreat, and more broadly, the TGC in influencing fjord geochemistry and ongoing coastal ocean fertilization, we sampled glacial sediment plumes and discrete

icebergs to understand the downstream geochemistry during tidewater glacier degradation. We present dissolved and particulate measurements of trace-metals in fjord surface seawater and discrete icebergs from two distinct fjords—both tidewater but with very different underlying bedrock geomorphology and retreat history—located in Southcentral Alaska. To investigate labile particulates in the nearshore environment, we used a chemical extraction to isolate the authigenic and adsorbed fraction of trace-metals as a proxy for the most bioavailable forms. Elemental ratios of particulate matter are leveraged to examine variations in the supply of trace-metal micronutrients during different modes of the TGC (e.g., glacier advance, stability, retreat). Up to this point, only sediment dynamics and soluble fluxes of macro- and micronutrients have been investigated, but not simultaneously.

## Results

### Study location and recent glacier change

Our two fjord study sites are located in Southcentral Alaska, located on the heavily glaciated Kenai Peninsula, which hosts the Harding Icefield and associated alpine outlet glaciers (Fig. 1). The two glaciers that are the focus of this study are the tidewater glaciers: Aialik Glacier (AG, 59.97°N, 149.84°W, 141.533 km²) and Northwestern Glacier (NWG, 59.88°N, 150.07°W, 50.299 km²) (Randolph Glacier Inventory v6.0). The most recent retreat of Aialik Glacier (AG) has exposed a moraine noted for subglacial meltwater discharge, while the terminus has remained in a stable position. Despite these changes, the net change was not documented as significant[6]. Northwestern Glacier (NWG) has changed dramatically throughout recent decades, with a net loss of 1.63 km² in the 1990s, which equates to a −1.59 km retreat of the terminus onto land. This is a continuation of a ~15 km retreat NWG has experienced since 1950. Factors which could affect their mass balance include sedimentation rates, fjord geometry, water depth, glacier geometry, ice speed, and calving rates. The bedrock geology is dominated by Eocene-Paleocene age (34 – 37 Ma) granitic rocks within the Harding Icefield region[42]. Metasedimentary rocks of the Valdez Group (Upper Cretaceous in age, 56 – 61 Ma) are present on the periphery of the fjords[43]. Given that both are outlet glaciers from the same icefield, we do not expect bedrock geology to be the dominant control on the geochemistry of the marine particulates measured in this study. We use AG as an idealized example of a stable tidewater configuration, while NWG, with its imminent transition to land-terminating following significant retreat, is our exemplar of a retreating configuration.

Area-altitude distributions can provide insight into meltwater production rates, as higher proportions of area located at lower elevation would result in higher rates of meltwater production[44]. Thus, AG produces more meltwater per melt-season than NWG, although the result of a larger catchment and accumulation at higher altitudes compensates for the current loss of ice mass at the glacier terminus. Satellite-based estimates of volume flux reveal AG has a glacial flux of 0.33 km³/yr and NWG has a flux of 0.07 km³/yr between 2014 to 2018[45]. Remote sensing at the heads of both fjords reveals large surface sediment plumes extending over the eastern flanks of the fjords and persisting for much of the inner basin (Fig. S1). There are distinct geomorphologies of the two tidewater glaciers, where AG has a broad terminus due to a lower grade slope and enters the fjord pressed against its terminal moraine. This is in contrast to NWG, which is narrow, steep, and has areas of exposed bedrock near the terminus, and is not in contact with a moraine. Waterfalls and icefalls were observed to be active in NWG, indicating many pathways for direct meltwater input to the surface waters of the fjord. To capture the spatial variability in seawater chemistry, a series of cross-fjord transects were sampled with varying distances from the glacier terminus, including one reference station seaward of the shallowest sill located within each fjord (Fig. 1b).

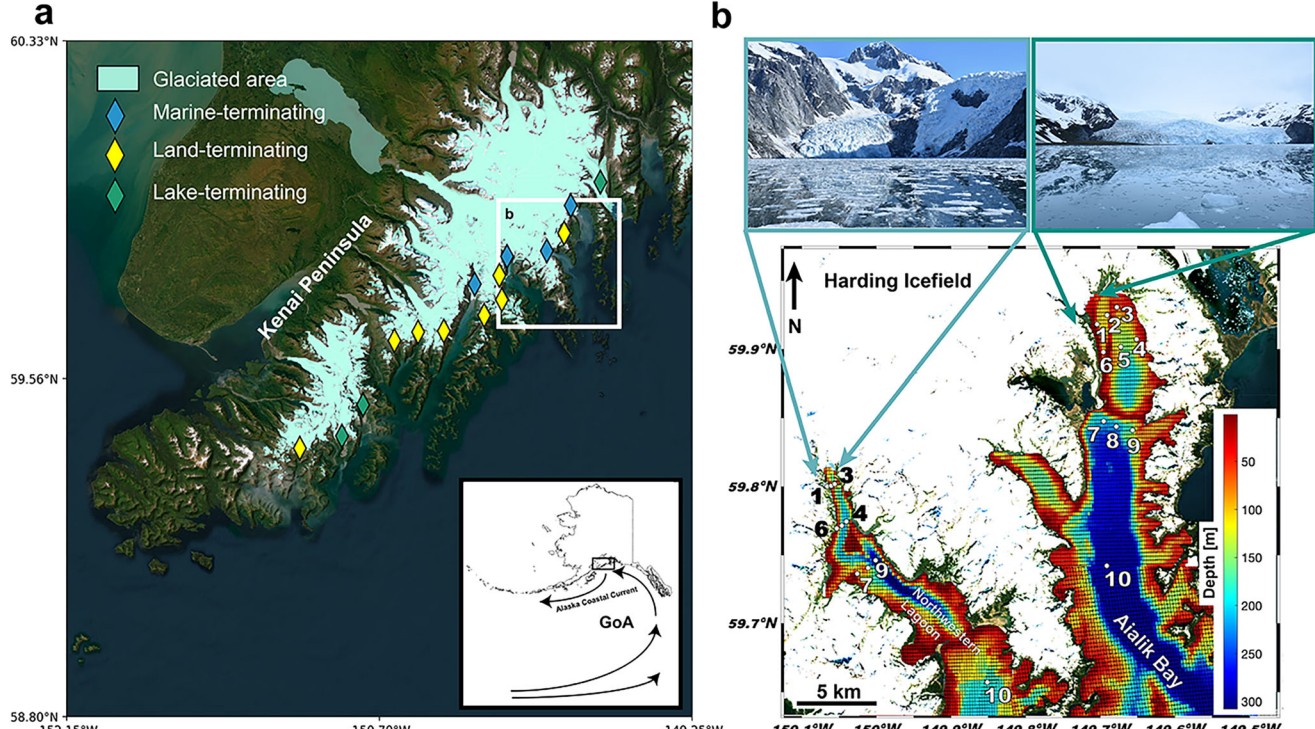

**Fig. 1 | Maps of the study location within a broader context of the region and approach to the study. a** The inset shows the location of the Kenai Peninsula with respect to the Alaska Coastal Current, which exchanges cross-shelf water with the Gulf of Alaska (GoA). The glaciated region is indicated by light blue shading, with outlet glaciers indicated by diamond symbols, where blue diamonds are marine-terminating, yellow are land-terminating, and green are lake-terminating. The fjords study region is indicated (**b**) with the prominent tidewater glaciers (Northwestern Glacier, Aialik Glacier) at the heads of the fjords and water column sampled stations as white points and numbered notation. Bathymetry is indicated by the colorbar. Maps (**a**) was made in ArcGIS, (**b**) is a composite image of Landsat satellite imagery (from May 9, 2022) and ocean bathymetry.

## Dissolved trace-metals and macronutrients

Macronutrients associated with offshore and subsurface water masses ($NO_3^-$, $PO_4^{3-}$) show an enrichment at station 10 in both fjords (Fig. S2). These relatively higher macronutrient concentrations generally persist along the eastern flanks of the fjords and decrease on the western flanks, indicating anticyclonic circulation and possible consumption of $PO_4^{3-}$ within the fjord. The anomalously high concentration of $NO_3^-$ at stations 6 (AG) and 7 (NWG) on the western flanks may be explained by two separate sources. In AG, at the terminus, subglacial meltwater entrains subsurface water masses within a rising turbulent buoyant plume. Therefore, high $NO_3^-$ at the surface here can be explained by a subsurface source within the sediment plume. The lowest salinities are found at station 6 (~26 ppt) and support this hypothesis. However, $PO_4^{3-}$ should also follow this pattern, as both macronutrients are enriched in subsurface waters. A high amount of oxides within AG, which are capable of scavenging $PO_4^{3-}$ from the water column due to their high surface area, could explain this discrepancy. Within NWG, we cannot explain surface concentrations by the same mechanism. Instead, we hypothesize that the anomalously high concentrations of both macronutrients at station 7 may be derived from local land-terminating glaciers, for which the Northwestern fjord contains two on the western flank (Fig. 1a). To some extent, the surface concentrations of $NO_3^-$ and $PO_4^{3-}$ would also be diluted by relatively macronutrient-poor glacial meltwaters and further scavenging loss of $PO_4^{3-}$ onto mineral surfaces[46]. In contrast, silicic acid ($Si(OH)_4$, or dSi) concentrations are more uniform throughout the fjord surface, with a small decreasing trend down-fjord, and do not show any trends with respect to circulation or potential for biological uptake (Fig. S2). There is a striking inter-fjord difference in dSi as well, as concentrations in AG (concentration range $0–2.5\,\mu M$) are much lower compared to NWG (concentration range $5–10\,\mu M$, Fig. 2).

The dFe concentrations in surface waters from AG ranged from 7 to 95 nM, but with large heterogeneity observed among stations (Fig. S3). There was no observable trend in dFe concentrations with respect to water column depth. The average for all inner-fjord stations ($n = 9$) was 32.77 nM, whereas station 10 located furthest from the fjord head representative of waters at the fjord mouth, had a concentration of 13.85 nM. Surface waters in NWG had greater concentrations of dFe compared to AG, with a range of 18.05 to a maximum of 216.61 nM, closest to the glacier terminus. The average concentration was 68.56 nM, more than double the average concentration found in AG. The rapid decay of dFe concentrations with distance from both glacier termini (Fig. S4) is also observed and modeled in a Greenlandic fjord[31] and over a similar salinity range in coastal Alaska[47], and demonstrates the importance of sampling the near-surface ($\sim 1\,m$), where meltwater may concentrate.

Surface waters of AG had dMn concentrations with a range of 14 to 34 nM, and a generally decreasing down-fjord concentration gradient, excluding station 6, which had the lowest concentration (8.81 nM). The average of all nine stations located within the inner-fjord at AG was 23.56 nM ($n = 9$). The station located at the fjord mouth seaward of the shallow sill had a concentration of 20.10 nM. In NWG, dMn concentrations were generally higher than AG and ranged from 25 to 58 nM, with the highest concentrations found at stations 7–9. The average for all nine stations within the NWG fjord was 31.55 nM. The station located at the fjord mouth, seaward of the shallow sill, had a concentration of 12.94 nM. Compared to dFe, the dMn concentrations do not attenuate as strongly with distance from the glacier terminus and across the salinity range (Figs. S3, S4), implying a more conservative behavior of dMn and its use as a possible tracer for recent glacial meltwater input of dFe to the ocean[48]. NWG had relatively greater concentrations of dMn at the surface compared to AG;

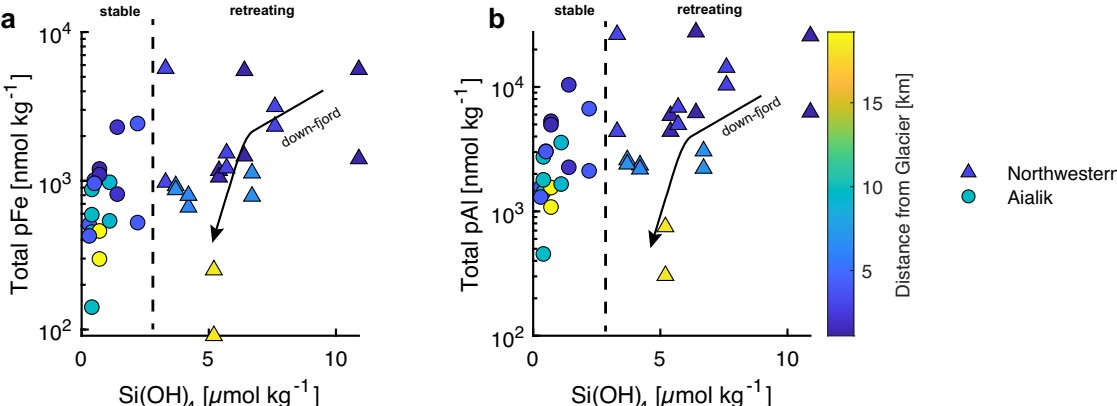

**Fig. 2 | Particulate metal and dissolved silica concentrations.** Concentrations of total particulate Fe (**a**) and Al (**b**) in suspended marine particles for Northwestern and Aialik fjords are plotted with dissolved Si concentrations. Symbols for the two fjords: Aialik fjord is circles, and Northwestern fjord is triangles. The colorbar displays the station distance from the glacier terminus.

however, the average concentrations of both fjords are of similar magnitude and comparable to concentrations found during the fall in surface seawater overlying the nearby continental shelf when glacial discharge is greatest in the region[49,50].

### Suspended particulate trace-metals

The largest contribution to particulate trace-metals were refractory, lithogenic detrital particles. The origin of these lithogenic particles and dSi is observed to be the glacier, as concentrations of total particulate Al and Fe (tracers for terrigenous inputs) and dSi generally decreased in the down-fjord direction consistent with particle sinking and dilution of meltwaters (Fig. 2). Concentrations of total particulate trace-metals (leached plus refractory components) decay exponentially with distance from the glacier terminus (Figs. S5, S6). Based on the summation of both particle size classes (<5 μm and >5 μm) to estimate total particulate Al, a relatively immobile element found in crustal aliminosilicates, we find that NWG produces more sediment than AG as average fjord surface concentrations are nearly twice as high (NWG: ~16000 nmol kg⁻¹, $n = 10$; AG: ~7900 nmol kg⁻¹, $n = 10$). This trend was also observed for Ti, Fe, Mn, and P (Table S1).

The particle leach provides insight into the labile fractions of Al, Ti, Fe, Mn, and P in suspended sediments ranging in size from 0.2 – 5 μm and greater than 5 μm. We refer to the labile fraction as $TM_{labile}$. $TM_{labile}$ showed a similar exponential decay of concentration with distance from the glacier terminus, suggesting a point source to the fjord surface. While Al, Ti, and P concentrations scaled with total particulates between the two fjords, summing both particle size classes ( < 5 μm and >5 μm), the average concentrations of labile particulate Fe (NWG: 480 ± 371 nmol kg⁻¹, $n = 10$; AG: 382 ± 288 nmol kg⁻¹, $n = 10$) and Mn (NWG: 16.8 ± 11.9 nmol kg⁻¹, $n = 10$; AG: 15.5 ± 9.2 nmol kg⁻¹, $n = 10$) were more uniform in the surface of both fjords.

The fraction of chemically labile total particulate trace-metals ($TM_{labile}$) is presented as averages and summarized in Fig. 3 for the elements studied here. We found that the two size fractions of particulates were similar in their fraction of $TM_{labile}$, excluding particulate P, which had a significantly higher fraction in the large particle size (NWG: $p = 0.001$; AG: $p = 0.000036$, paired $t$-test). Between fjords, we found that Fe and Mn had higher fractions of $TM_{labile}$ in AG (>5 μm: 0.15 ± 0.02 Fe, 0.25 ± 0.07 Mn; <5 μm: 0.15 ± 0.01 Fe, 0.26 ± 0.03 Mn) compared to NWG (>5 μm: 0.13 ± 0.02 Fe, 0.15 ± 0.02 Mn; <5 μm: 0.11 ± 0.01 Fe, 0.13 ± 0.03 Mn), independent of particle size.

### Iceberg-laden sediments and cobbles

Glacial ice contains lower concentrations of total particulate matter and trace elements on average compared to our fjord seawater particulate samples, highlighting the significant role that subglacial discharge plays in sediment and elemental transport, whereas most calved glacial ice (icebergs) were relatively clean and contained little entrained particulate matter. The basal ice, which is in contact with the bedrock, contains concentrated amounts of particulate matter[19]. The $TM_{labile}$ component was measured in iceberg-laden sediments in two size classes: >10 μm and <10 μm particles. Lower overall labilities were measured in the >10 μm fraction, compared to particulates from glacial sediment plumes for the most mobile elements, but similar to bedrock samples (0.016 ± 0.005 Al, 0.009 ± 0.006 Ti, 0.031 ± 0.015 Fe, 0.027 ± 0.015 Mn, 0.121 ± 0.035 P, $n = 5$). For the smaller particle fraction (<10 μm), chemical labilities were slightly higher and more comparable to glacial sediment plumes (0.031 ± 0.020 Al, 0.037 ± 0.041 Ti, 0.051 ± 0.023 Fe, 0.057 ± 0.038 Mn, 0.15 ± 0.097 P, $n = 5$).

There exists no published trace element geochemical data for the bedrock underlying the Harding Ice Field in the Kenai Peninsula. Therefore, cobbles isolated from two icebergs were chosen to be representative samples of the rock weathered by outlet glaciers from the Harding Ice Field. Similar to the large sediments entrained in icebergs, cobbles had overall lower $TM_{labile}$ fractions compared to sediments in the fjord water column (0.015 Al, 0.007 ± 0.002 Ti, 0.030 ± 0.005 Fe, 0.11 ± 0.01 Mn, 0.28 ± 0.01 P, $n = 2$). An in-depth analysis of rare earth elements in suspended particles and cobbles shows minor geochemical differences between fjords, which do not drive the observed trends in elemental concentrations discussed below (see Supplemental Discussion for more information).

## Discussion

### Geochemical tracers of weathering reflect a glacier in transition

The transport and residence time of meltwaters in alpine glaciers is generally short, ranging from 1 to 100 h, and in the case of tidewater glaciers, which terminate in the ocean, they provide direct route of input of glacial weathering products to the coastal ocean[51,52]. By comparison, the extended residence time of meltwaters in subglacial beds and cavities of ice sheets of Greenland and Antarctica (weeks to >10 kyr) greatly decouples the supply of dissolved and labile particulate trace-metals to the ocean[53,54]. Seasonal and longer-term reorganization of subglacial drainage channels has a large impact on the residence times of meltwater and sediments, with both shown to greatly decrease over the course of a melt season in alpine systems[41]. However, it is not well constrained how glacial retreats impact sediment transport or the geochemistry of stored sediments; for example, tidewater glacial retreat may result in increased transport of exposed sediment through outwash and fluvial environments.

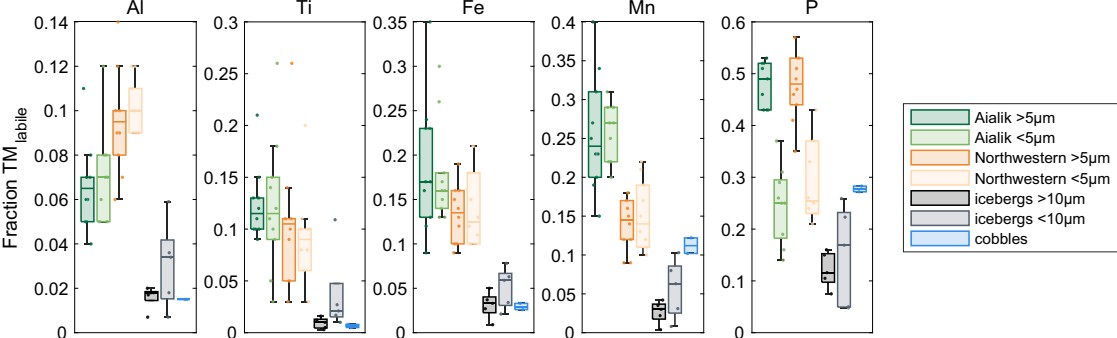

**Fig. 3 | Chemical lability of fjord particulate matter.** Mean labile fraction of particulate trace metal (TM$_{labile}$) within each fjord and in two particle size fractions, where each plot is grouped by element with increasing element mobility from left to right (Al to P). The inner two quartiles are plotted as vertical boxes with the 95% confidence interval plotted as whiskers. Individual data points are shown as dots.

Colors of each box reflect the sample type: dark green are coarse sediments (>5 µm) in Aialik fjord; light green are fine sediments in Aialik fjord; dark orange are coarse sediments in Northwestern fjord; light orange are fine sediments in Northwestern fjord; dark gray are coarse (>10 µm) iceberg-laden sediments; light gray are fine (<10 µm) iceberg-laden sediments; and blue are cobbles.

Silicic acid (dSi) is a useful tracer for gauging the level of intense glacial weathering of silicate bedrock[55]. Greater dSi concentrations in meltwaters implies heightened rates of chemical weathering of silicate bedrock through increased contact time between meltwater and bedrock (i.e., storage)[56]. NWG has more intense weathering activity at the fjord surface (~1 m depth), indicated by high concentrations of dSi, whereas AG fjord had consistently lower dSi concentrations. The trends in observed dSi are not related to the amount of meltwater generated in this case, as AG produces more meltwater compared to NWG by an approximate factor of three[45].

Observations of dFe and dMn support that comminuted sediments beneath NWG experience greater chemical weathering intensity than those beneath AG, with on-average greater concentrations of dFe and dMn in NWG fjord surface waters (31.55 ± 10.65 nM dMn, 68.56 ± 61.55 nM dFe) than AG (23.56 ± 8.26 nM dMn, 32.77 ± 26.48 nM dFe). DFe appears to be positively correlated with dSi within NWG, but not AG. Dissolved Fe and Mn are more strongly correlated with salinity within NWG (r = −0.66, −0.46, respectively) compared to AG (r = 0.13, −0.08, respectively), however large variability exists between stations that cannot be explained by solely meltwater addition to the fjord surface. These modest correlations support the notion that meltwaters within NWG supply higher rates of dissolved trace-metals due to enhanced chemical weathering occurring beneath NWG. In a retreating scenario, as observed at NWG, higher rates of chemical weathering are expected. Alaskan glaciers are documented as having some of the highest cation denudation rates among global alpine glaciers, resulting from climate warming-induced increases in meltwater discharge[57,58].

As subglacial meltwater flushes through bedrock and glacial till surfaces, chemical weathering fluxes are a byproduct of water-rock interaction time[56]. Mechanical weathering primarily occurs during periods of glacial advance[59], and produces high surface area sediments which when exposed to water for prolonged periods of time (e.g., chemically weathered) will result in higher dissolved metal concentrations. Reductions in the thickness of NWG[45], and therefore, bed-normal stress and basal friction, and increased deformation of a layer of subglacial sediments also limit direct mechanical weathering of bedrock[60]. We expect dSi and dissolved trace-metals to be correlated with salinity because of significant meltwater input within NWG. However, dSi is not correlated with salinity (r = −0.40) or dMn (r = −0.25) and is correlated with dFe (r = 0.76). No correlations exist between dissolved Fe and Mn and dSi within AG. Elevated dFe in subglacial meltwaters (10–100 s nM) likely indicates recent discharge to the ocean and incomplete equilibration with particle surfaces and organic ligands. No correlation exists between dFe and TM$_{labile}$ in AG (r = −0.007), but a strong correlation in NWG (r = 0.71) suggests an equilibrium has been established through dissolved-particle

exchange[61], which occurs during longer storage times of sediment-rich meltwaters beneath NWG.

The enrichment of Mn relative to crustal Fe in both fjords is due to an excess of labile Mn in plumes (Fig. 4e,f). The Mn-to-Fe ratios in the dissolved fraction (AG: 1.1 ± 0.7; NWG: 0.8 ± 0.6) are considerably higher than the total and labile particulate Mn-to-Fe ratios (AG: 0.05 ± 0.02; NWG: 0.04 ± 0.01) (Fig. S3), illustrating the importance of slower exchange of Mn with particles in the sunlit surface ocean[62,63], compared to Fe, which rapidly equilibrates with TM$_{labile}$[64]. The coupling of Mn and Fe in glacial systems, likely due to similar redox and scavenging processes in the subglacial environment, is evidenced by the similarity in labile particulate Mn-to-Fe ratios of these two fjords. However, marine surficial sediments, known to have elevated Mn-to-Fe ratios due to anaerobic reductive dissolution processes, might drive higher ratios to occur in the AG water column[29] (Fig. 4f). These sediments could be resuspended during glacial scouring of the seafloor and remain suspended, as subglacial lithogenic particles are lost to sedimentation in the inner fjord[21].

We rule out the possibility of significant differences in biogenics between the fjords since similar concentrations of labile particulate P were observed in both fjords. The labile fraction is likely to contain phytoplankton cells in fjord surface waters, in addition to PO$_4^{3-}$ scavenged onto authigenic oxide minerals. Therefore, similarity in the concentrations of labile and total particulate P of both fjords allows us to infer that there are similar abundances of phytoplankton cells[65,66]. However, P was the only element which displayed a significant difference in the average lability of particulates between size fractions (<5 µm: 26 ± 7% and 24 ± 8% for AG and NWG, respectively; >5 µm: 49 ± 4% and 47 ± 7% for AG and NWG, respectively). This pattern is likely driven by the presence of some biogenic particles, which would be retained on the larger pore-size filter. The leach conditions used in this study are known to access both labile authigenic phases of Fe and Mn, as well as biogenics, due to the exposure to heating. Therefore, we rule out the presence of silicifying phytoplankton and uptake of dSi in AG as an important factor controlling surface concentrations. Additionally, the intense turbidity of the inner fjord is expected to be an important factor limiting phytoplankton productivity by reducing photosynthetically active irradiation of the fjord surface[7].

### Glacial setting drives downstream sediment geochemistry

The TGC shapes how sediments are delivered and transformed within fjords by modulating sediment plume dynamics, particle transport, and water-rock interaction. At AG, a stable tidewater glacier with a broad terminus and low angle of intersection with the ocean, subglacial meltwaters rise as a buoyant subsurface plume which then entrains marine sediments and undergoes density-driven particle

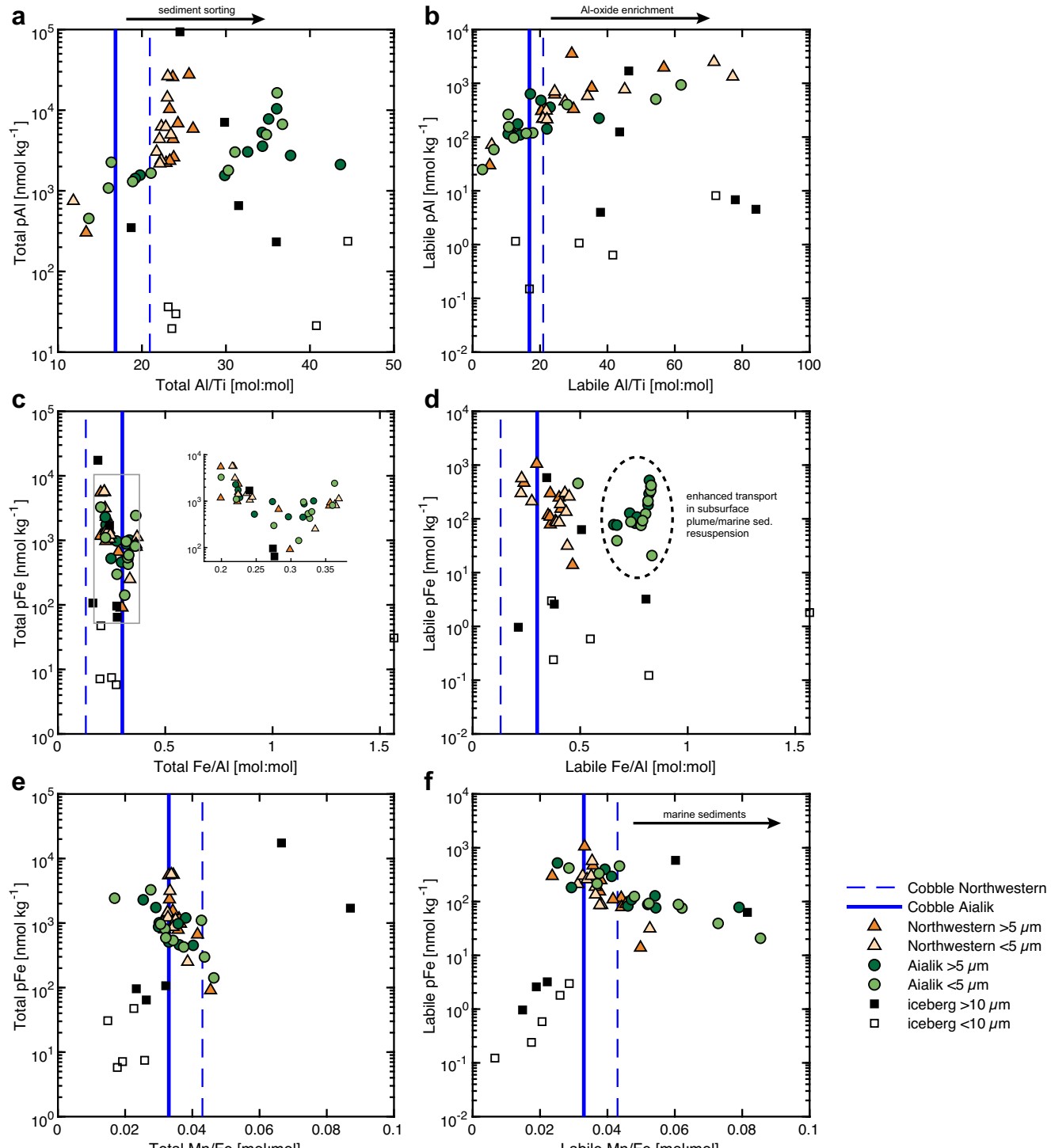

**Fig. 4 | Elemental ratios of fjord particulate matter.** Concentrations of total and labile particulate Al (**a**, **b**) and Fe (**c**–**f**) in suspended marine particles for Aialik and Northwestern fjords and iceberg-laden sediments are plotted with Al-to-Ti (**a**, **b**) ratios, Fe-to-Al (**c**, **d**) and Mn-to-Fe (**e**, **f**) ratios. Symbols for the two sites and particulate matter: Aialik fjord marine particles are triangles with the size fractions following the color scheme in Fig. 3; Northwestern fjord marine particles are circles with the size fractions following the color scheme in Fig. 3; coarse (>10 μm) iceberg-laden sediments are black squares; and fine (<10 μm) iceberg-laden sediments are white squares. For the iceberg-rafted cobbles (upper continental crust), elemental ratios are displayed as blue vertical lines in each plot: the solid blue line is the cobble from Aialik fjord, and the dashed blue line is the cobble from Northwestern fjord.

sorting during ascent. This process preferentially removes coarse, Ti-rich grains and enhances the relative contribution of fine particles, reflected in higher observed detrital and total Al-to-Ti ratios (Fig. 4a) and a greater proportion of fine (<5 μm) particles (see Supplemental Discussion). The dominance of sorting indicates that mechanical weathering and transport processes largely govern the AG sediment

geochemistry signal. In contrast, NWG subglacial meltwater discharges directly at the fjord surface, minimizing opportunities for fluvial sorting and yielding lower detrital and total Al-to-Ti ratios (Fig. 4a) with lower fine-particle contributions (Table S2). Instead, sediments here exhibit markedly higher labile Al-to-Ti ratios (Fig. 4b), signaling the presence of secondary Al-oxides formed through intense chemical

alteration of aluminosilicates. This enhanced chemical weathering is consistent with prolonged subglacial residence times and extensive water-rock interaction beneath the retreating tidewater glacier. Together, these contrasts demonstrate how the TGC regulates the balance between mechanical and chemical weathering: buoyant subsurface plumes (AG) promote physical sorting and export of fine particles, while surface discharge systems (NWG) favor strong chemical alteration during interactions in the water column.

We present additional evidence in support of increased chemical weathering in NWG associated with the transport of previously lithified sediments. Overall, there are no differences in chemical lability between our two particulate size fractions studied within each fjord, suggesting that over these short distances, suspended sediments remain well-mixed. We found that AG particulate matter is systematically more enriched in labile Fe relative to crustal Al (Fig. 4d) compared to samples from NWG. Whereas scavenging of Al-oxides is an important process in both fjords, indicated by elevated labile Al relative to crustal Ti[67] (Fig. 4b), particulate matter within AG is highly enriched in authigenic and scavenged forms of redox-sensitive Fe and Mn (Fig. 3). This can be attributed to the presence of a buoyant subglacial plume, which entrains subglacial and marine sediment particles, both sources known to contain high concentrations of authigenic Fe and Mn- oxides formed at the seawater oxidative front[28,29,68]. Deposition and accumulation of subglacial sediments is apparent at the AG terminus, consistent with the formation of a sill during tidewater glacier stability[13]. The elevated Mn relative to crustal Fe in all particulates reflects biogeochemical coupling of these elements and recent input of rich Fe sources[50].

The concentrations of total and labile particulate Fe and Al decrease with greater distance from the head of the fjord. Despite a nearly two-fold difference in total concentrations of particulate trace-metals, we attribute the similar labile particulate Fe concentrations to an intrafjord difference in the chemical labilities of Fe and Mn contained within the glacial sediment plumes. Of all of the metrics measured, the only significant differences between the two fjords studied here were the average chemical labilities of particulate Mn and Fe. Interestingly, only particulate P was, on-average, more labile in the large size fraction, compared to the small particle size fraction. These observations were consistent between the two fjords and may be linked to a similar amount of biogenic particulate matter present in both fjords (Fig. S5). While ocean hydrodynamics may play a role in the retention of sediments within the inner fjords through restricting lateral advection, recirculation is not expected to occur in narrow fjords[69] and therefore cannot explain the lower salinities observed and higher total particulate trace-metals in NWG.

We hypothesize that the qualitative differences between particulate matter sampled at each fjord are related to the residence time of sediments in the subglacial hydrologic system. The differences may indicate variations in the residence and transport time of sediments that were freshly eroded from silicate bedrock at the equilibrium line altitude (exhausted sediment scenario) or transport-limited sediments stored at higher elevation and eventually transported towards the fjord (Fig. 5). The approximate location of maximum erosion typically occurs at or just below the equilibrium line, which is supported by modeling frameworks that incorporate glacial hydrology, erosion, and sediment transport[13,70]. Despite a decrease in the catchment area, models indicate that retreating glaciers produce more sediment that was previously inaccessible through enhanced melting and hydrologic transport of subglacial till layers located at higher elevation[70,71]. In the case of the retreating NWG, the position of maximum erosion is unstable and has likely shifted towards higher elevation, where access to previously stored sediments is now possible[72]. Glacier flow velocities at NWG are highest at higher elevation, relative to AG, which is characteristic of land-terminating glaciers on the Kenai Peninsula[45]. These previously transport-limited sediments have experienced long residence times in the subglacial environment and are extensively chemically weathered, resulting in the loss of relatively immobile elements, like Al, Ti, and Fe[34,73]. These chemical weathering products are derived from dissolution of prevalent mineral phases, such as Fe-, Si-, and Al-rich biotite[58], which comprises up to 30% of the minerals of the Harding Icefield batholith[74]. The stored sediments, leached of labile metals during intense chemical weathering, retain a refractory signature that is subsequently transported to the fjord surface as observed in NWG.

A recent compilation of elemental data of exposed subglacial sediments from 19 retreating, land-terminating glaciers across the Arctic[73] lends additional support of this process by demonstrating low total particulate Al-to-Ti ratios ($5.2 \pm 3.3$ mol:mol) in sediments with highly variable bedrock composition. While chemical speciation was not investigated, the total particulate Fe-to-Al ratios are elevated in these sediments, even higher than what was found in AG. In proglacial environments, it is possible that remobilized Fe within newly exposed subglacial sediments results from microbial degradation of ancient organic matter[73,75–77]. The interaction of Fe and organic matter in proglacial environments (in front of or adjacent to the glacier) is likely to affect the fate of Fe. Proglacial processes such as metal remobilization via microbial degradation may drive sediments towards higher Fe-to-Al ratios through the production of labile Fe secondary phases, however, they are likely not transported to the coastal ocean, as winds are landward throughout the spring and summer[78] and proglacial flood events from the drainage of meltwater lakes typically occur in September and October[79] and thus do not contribute to particulate Fe and Mn supplied to the coastal (macronutrient-limited) and off-shelf (Fe-limited) regions during phytoplankton peak productivity during the Spring bloom in May[80].

Compositionally, the NWG sediment plume is likely composed of aged secondary minerals such as clays, which reflect a heavily weathered bedrock source signature. The prominent tidewater terminus of AG promotes a direct injection of a subsurface buoyant plume of labile Fe and Mn to fjord waters, followed by loss of some coarse sediments as the buoyant plume rises to the fjord surface. This labile material, likely composed of primary minerals and freshly precipitated secondary minerals (e.g., amorphous Fe oxy(hydr)oxides), is derived from recent erosion of bedrock. Consequently, freshly eroded bedrock under AG, generation of high surface area nanoparticles, precipitation of reduced forms of metals as authigenics, and entrainment of seawater drive the labile fraction to be highly enriched in the particulate phase[81].

Labile particulate trace-metals are characterized by a higher fertilizing potential compared to total particulate trace-metals, which include a large refractory component that is not solubilized under the leach conditions applied in this study. Dissolved Fe and Mn concentrations—the most bioavailable form we observed—remain two orders of magnitude higher than open-ocean concentrations observed beyond the shallowest sill. Dissolved metals are subject to processes which strongly attenuate concentrations down-fjord, such as particle adsorption. Considering this pool is orders of magnitude lower than labile particulate trace-metals, we limit our discussion to the particulate pool, which dominates Fe and Mn speciation.

## Sediment plumes chronicle subglacial weathering

The Kenai Peninsula fjord system in Southcentral Alaska hosts fifteen alpine glaciers adjacent to the continental shelf, comprising four tidewater, eight land-terminating, and three lake-terminating (Fig. 1). This work builds on our understanding of the degradation history of tidewater glaciers and how changes in the geomorphology through time impact the geochemistry of the coastal ocean. To address this question, we surveyed the Al, Ti, Fe, Mn, and P lability of particulate matter suspended in the surface waters of two geomorphologically distinct fjords, at two distinct stages of the TGC: stable and retreating.

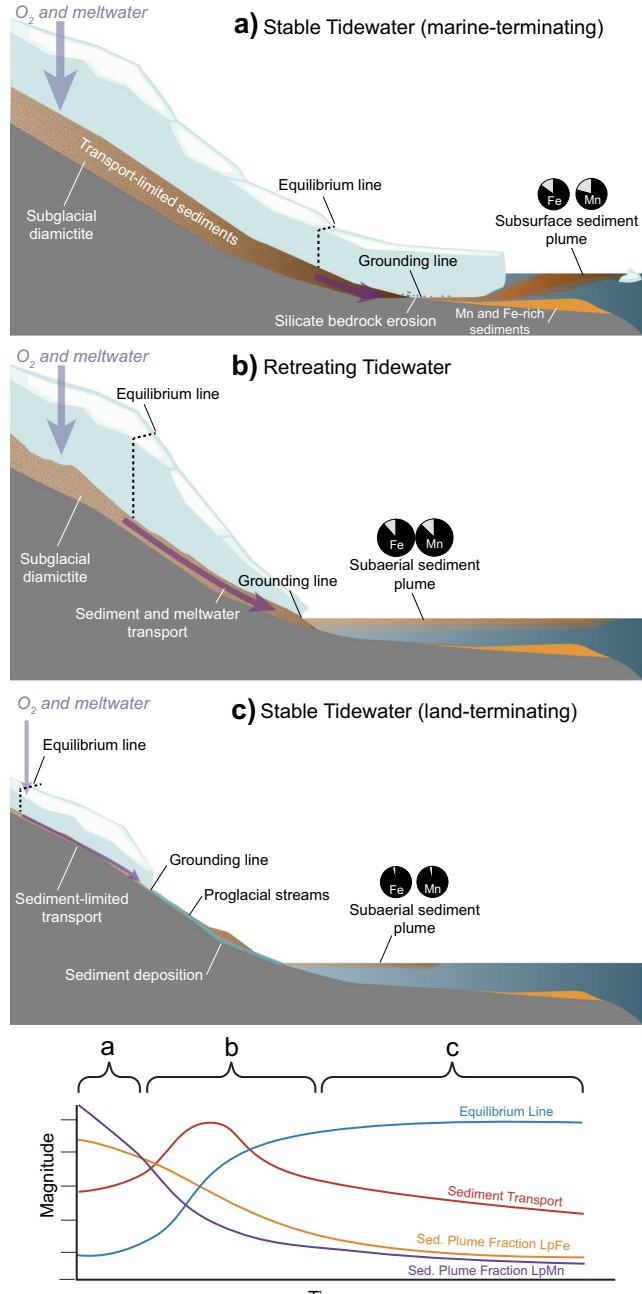

**Fig. 5 | Schematic model for the transport of sediments from glaciomarine environments and associated sediment plume geochemistry.** A stable tidewater (**a**) erodes fresh bedrock where anoxic subglacial meltwaters mix with fjord seawater at the grounding line, producing highly labile particulate matter enriched in Fe and Mn co-precipitated at the ice-ocean boundary. A stable tidewater stores subglacial sediments (transport-limited) above the point of maximum erosion, and relatively immobile elements (Al, Fe) are not retained. In the retreating state (**b**), the point of maximum erosion moves to a higher altitude. Stored subglacial sediments, lacking labile metals, can now be transported directly to the glacier surface as a subaerial plume. Once land-terminating (**c**), sediments are exhausted at the grounding line and particulate matter generated from erosion is deposited in proglacial streams, reducing the sediment flux to the ocean. Sediment plume fractional TM$_{labile}$ are displayed as pie-charts (gray = labile, black = refractory), where reductions in lability of Fe and Mn are observed between AG (**a**) and NWG (**b**), and (**c**) from the mouth of the Copper River in Lippiatt et al. (2010)[36] for Fe (LpFe ~4% of TpFe) and scaled for Mn.

The proximity of our study sites to one another allowed us to infer the effects of tidewater glacier retreat on the chemical lability of sediments transported to the ocean by controlling for the geology of the eroded bedrock source.

Fjord sediments have the potential to provide high-resolution sedimentary and geochemical records of past environmental and climate change, such as meltwater discharge or glacial mass balance. In polar regions, flucuations in trace-metal lability have been interpreted as diagenetic alterations during benthic cycling related to oxygen and organic matter availability on years to decades timescales[21,82]. We contend that fluctuations in the chemical lability of detrital sedimentary material from tidewater glaciers may also arise as a result of its geomorphology (i.e., advance and retreat cycle), which is expected to change on century to millennial timescales. This does not preclude changes in meltwater production and sediment transport associated with changes in atmospheric moisture and sea surface temperatures[83], as seasonal and annual changes in sediment discharge are well documented[70,84,85]. Future climate warming could have an increasingly significant effect on erosion rates and sediment production[14,86], not captured by the TGC. Eventually, a complete transition onto land slows sediment transport to the ocean, as tidewater glaciers experience reductions in catchment area. In the shorter term, accurate knowledge of how glacier retreat will increase sediment discharge to the ocean would require extensive mapping of stored subglacial till[70,87] and an expanded geochemical sampling campaign across fjord systems within a single icefield. We have shown trace-metal ratios and chemical lability evolve within a short distance of the glacier terminus, suggesting the far-field fertilization potential of particulate matter derived from glacial weathering is sensitive to tidewater geomorphology, which influences glacial degradation and sediment storage time.

## Methods

### Description of glaciomarine sample sites

Both tidewater glaciers experienced early change across the 1990s, primarily through decreases in the lower glacier area (AG and NWG) and glacier length (NWG). Since the 2000s, metrics such as glacier area and length have stabilized, but there may be compensation in other glacial health metrics, such as glacial thinning. As continued melt is anticipated, it is important to note that these metrics do not necessarily indicate changes in mass balance, but given that both glaciers experience similar climate, we would expect regional warming to result in glacial thinning.

Both fjords contain concentrated icebergs and brash ice within the inner basins, which were concentrated on the northwest flank of the fjords. NWG had noticeably more ice concentrated in the inner basin, possibly due to the long residence time of seawater in the narrowest section of the fjord (Fig. S7). Air temperatures were anomalously warm for the sampling period, causing much of the floating ice to melt, which is reflected in lower salinities and colder temperatures at the surface. Salinities varied down-fjord, where AG had a salinity minimum of 28.5 ppt (practical salinity units) at station 3 and a maximum at station 9 of 32.5 ppt. NWG had lower overall salinities at the surface, with a down-fjord gradient of 25 ppt at station 1 to 32 ppt at station 10. Surface water temperatures had a down-fjord gradient and ranged from 1.2 to 9.8 °C in AG and 0.2 to 10.7 °C in NWG.

### Sampling of seawater, sediment plumes, and icebergs

Fieldwork was conducted aboard a small craft (National Park Service – Kenai Fjords National Park) over two days in May 2022. Sampling stations in each fjord were designed as three transects with increasing distance from the glacier terminus and a reference station located seaward of the shallow sill (Fig. 1). All sampling gear and methods for the collection of seawater, suspended marine particulate matter, and icebergs were to the standards of trace-metal clean research (Cutter et al.[88]). Seawater was collected using acid-cleaned Teflon tubing

attached to an extended aluminum boat hook and weighed down to -1 m below the surface using an amsteel line attached to a coated 2 kg weight located -2 m below the surface. In-line filtration using stacked acid-cleaned 5 and 0.4 μm polycarbonate filters collected marine particulates, while the filtrate was collected into acid-cleaned 30 mL LDPE bottles and acidified to pH 1.8 with HCl (optima grade). Pieces of icebergs were collected from above the freeboard to minimize the influence of seawater, handled with polyethylene gloves, and placed into large plastic bags. Icebergs were maintained frozen (−20 °C) until processing in a clean room.

Protocols were developed for sampling of particulate matter within icebergs. Each piece of ice was removed from the −20 °C freezer, and the surface allowed to warm (turning to a glossy sheen). This step prevents refreezing during the decontamination step, as follows: Approximately 10% (by mass) of the outermost ice is removed by melting with ultrapure MilliQ (18.2 MΩ) water. The remaining 90% of ice is placed into an acid-cleaned Teflon bag, sealed, and melted at room temperature. Once melted, the liquid is homogenized and pressure-filtered onto stacked acid-cleaned PTFE (Sartorius) filters of the sizes 10 μm and 0.2 μm. The filters and collected particulate matter were left to dry overnight in a HEPA clean-air laminar flow bench.

Cleaning procedures differed for the filters, sampling equipment, and Teflon vials. All sampling equipment, including Teflon tubing, pump tubing, filter housing, and filter slides, was soaked overnight in an acidic detergent (Citranox) and heated to 60 °C. Next, materials were either placed into a 3 N $HNO_3$ (trace metal grade) bath or filled with this acid and left to soak for one week at room temperature. Materials were then placed into a 3 N HCl bath (trace metal grade) or filled with acid for another week at room temperature. Following each step, materials were rinsed five times with ultrapure MilliQ water. Vials followed a similar cleaning protocol to sampling equipment, but were heated to 110 °C, and received an additional cleaning step with concentrated $HNO_3$ (optima grade) and trace HF (optima grade). Polycarbonate filters were rinsed with methanol, and then soaked overnight in 6 N HCl (optima grade), before being placed into pH 1.8 ultrapure deionized water until use. Teflon filters were cleaned with 6 N HCl (optima grade) at 110 °C overnight and then aqua regia (3:1 v/v HCl and $HNO_3$ mixture, optima grade) overnight at 110 °C. Finally, concentrated $HNO_3$ (optima grade) is added to the Teflon filters with a trace amount of HF (optima grade) and allowed to reflux for 3 more nights at 110 °C. All filter types were rinsed five times with ultrapure MilliQ water between steps.

Seawater dissolved Fe and Mn concentrations were prepared on a seaFAST pre-concentration unit (ESI) in an offline configuration with matrix removal and measured on an Element 2 (Thermo) Inductively coupled plasma-mass spectrometer (ICP-MS) at the University of Washington in December 2023 using external calibration: 0, 0.5, 1, 2, 5, 15 ppb of Solution 2 multi-element standard in 5% v/v $HNO_3$ (Claritas PPT®). The methods are according to published methods in Rapp et al. (2017)[89]. Standards were processed the same as samples. The accuracy and precision of the method were checked by analysis of the certified Geotraces Surface Coastal Pacific seawater reference material (GSC: [dMn] = 2.84 nM, [dFe] = 1.77 nM) and the certified reference material, NASS-7 ([dMn] = 13.65 ± 0.38 nM, [dFe] = 6.18 ± 0.27, $n$ = 4). These values are within the uncertainty of the consensus values (GSC: [dMn] = 2.180 ± 0.075 nM, [dFe] = 1.535 ± 0.115; NASS-7: [dMn] = 13.64 ± 1.09 nM, [dFe] = 6.27 ± 0.46 nM).

Approximately 30-35 mL of in situ water column filtrate from a 0.2-μm capsule filter was dispensed into 50 mL polypropylene centrifuge tubes and frozen at −20 °C until on-shore autoanalyzer analysis. Each tube was rinsed with sample three times before filling. Nutrients were analysed on a QuAAtro continuous segmented flow autoanalyzer (SEAL Analytical) as described by the CalCOFI methods manual (https://ccelter.ucsd.edu/cce-calcofi-methods-manual/). Total oxidized nitrogen (nitrate reduced to nitrite), solely nitrite, and the efficiency of reduction of nitrate to nitrite were assessed to calculate solely the nitrate concentration. Reference materials for nutrients in seawater (KANSO Co., LTD.) were included in the run for quality control.

## Sequential extraction of particulate trace-metals

Following a protocol based on Berger et al. (2008) and modifications applied by Milne et al. (2017), filters were subjected to a sequential leach and digest procedure. The acetic acid (Hac, 25% v/v at pH 2, optima grade) with hydroxalamine hydrochloride (0.02 M, 99.995% trace-metals basis) offers mild conditions for the solubilization of the most chemically labile metal precipitates (i.e., ferrihydrite (FeO(OH)), manganese oxides ($MnO_2$), as well as metals associated with biogenics (termed $TM_{labile}$). Briefly, the filters were folded loosely into eighths and placed into an acid-cleaned Teflon vial, and 2 mL of leach solution was added to the vial, capped, and immediately placed into an oven at 90 °C for 30 min. Once the heating time had elapsed, the samples were allowed to cool to room temperature over the next 90 min. Once two hours of contact time with the solution and filter were reached, the filter was removed and placed into a second 15 mL acid-cleaned Teflon vial (Savillex). The leached portion was centrifuged at 13000 × $g$ for 15 min, and 1.5 mL of the leachate was removed and placed into acid-cleaned 7 mL PFA vials (Savillex), dried at 110 °C, re-solubilized in 100 μL concentrated nitric acid, dried, and taken up in 3 mL of 2% v/v $HNO_3$ spiked with 10 ppb In (ICP-MS solution). The leached fraction was refluxed for 8 hrs or overnight in the ICP-MS solution at 110 °C with the vials capped.

The filters were subsequently exposed to a digestion protocol in which 1.5 mL mixture of $HNO_3$ (-86% v/v, optima grade) and HF ( -14% v/v, optima grade) was added to each vial[61]. The vial was tightly capped and refluxed overnight on a hotplate at 110 °C. The next day, vials were allowed to cool, and any condensation on the vial walls and cap was collected on the vial bottom before uncapping and drying down until -10 μL droplet of acid mixture remained, at which point it was necessary to stick the filter against the vial wall using acid-cleaned Teflon tweezers. Next, 1 mL of aqua regia was added directly to the droplet, the vial was tightly capped, and refluxed overnight on a hotplate at 110 °C. The following day, the acid was cooled and collected prior to drying down to a -10 μL droplet, and 1 mL of 6 N HCl (optima grade) was added. The vials were capped and refluxed overnight at 110 °C. A final dry-down step was conducted, and the filter was removed. The following day, 5 mL of 2% v/v $HNO_3$ (optima grade) solution containing 10 ppb In was added. The solution was allowed to redissolve any residue for at least 8 h of refluxing at 110 °C on a hotplate. Samples were diluted 50 – 200 x prior to analysis. The efficiency and accuracy of these methods are examined using community standards, namely GSP-1 (USGS) and AZTD (Powder Technology, Inc.). The results of the accuracy check are presented in Table 1.

Particle leach and digest solutions were analyzed on an iCAP Qc (Thermo) inductively coupled plasma mass spectrometer at the Irvine Materials Research Institute at the University of California, Irvine. A full suite of nine elements were analyzed, and so we consider major lithogenic elements (aluminum (Al), titanium (Ti), Fe, Mn) in addition to other bioessential elements (phosphorus (P)). All standards and samples were introduced into the plasma as 2% v/v $HNO_3$ solutions with 10 ng $mL^{-1}$ In as an internal standard to monitor instrumental drift and matrix effects. The internal standard was prepared volumetrically via serial dilution. Multi-element calibration standards (0.1, 1, 10, 100, 1000 ppb) were prepared by serial dilution of CMS and 71-A (Inorganic Ventures) standards into 2% v/v $HNO_3$. Following the calibration curve, several blanks were run as rinse solutions to confirm a low background prior to analyzing samples. Filter blanks (including acid and vial blanks) average corrections were: 1.3% Al, 5.7% Ti, 0.45% Fe, 11% Mn, and 19% P for particle leaches; 0.15% Al, 1% Ti, 0.1% Fe, 2.8% Mn, and 11.9% P for particle digests.

**Table 1 | Results of the accuracy check for certified geologic reference material GSP-1 (granidiorite powder) and AZTD (Arizona Test Dust, ISO 12103-1 ultra fine, Powder Technologies, Inc.)**

| Element | GSP-1 | Certified | Accuracy | AZTD | Shelley et al. (2015) | Accuracy |
|---|---|---|---|---|---|---|
| | [ppm] | [ppm] | [%] | [ppm] | [ppm] | [%] |
| Al | 90996 | 81959 | 11 | 62754 | 61100 | 3 |
| Ti | 4443 | 4538 | −2 | 3662 | 3220 | 14 |
| Fe | 34221 | 34300 | 0 | 33351 | 33600 | −1 |
| Mn | 317 | 320 | −1 | 639 | 555 | 15 |
| P | 1171 | 1300 | −10 | nd | nd | nd |

GSP-1 certified elemental concentrations are from Raczek et al., 2001[90]; AZTD elemental concentrations were published in Shelley et al., 2015[91].

The filtered particles were standardized based on the volume of seawater filtered; therefore, the elemental concentrations reported in this study are representative of glacially derived metals transported to the marine environment per unit volume of water. Factors that may influence the concentrations of elements in the particulate phase include: 1) the amount of sediment transported through physical weathering; 2) the subglacial chemical weathering intensity; 3) initial bedrock composition; 4) transport system and distance from glacial terminus to fjord environment. All these parameters, excluding bedrock composition, can be strongly influenced by glacial setting and geomorphology, and gauging this influence is a primary goal of this study. It is important to note that the samples presented in this study represent a snapshot in time and are likely to fluctuate over the course of the melt season.

## Data availability
Source data is provided with this paper.

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

## Acknowledgments

We thank Kenai Fjords National Park personnel, Deborah Kurtz (science coordinator), and boat captain Nathan Bawtenheimer for their support of this research. Two volunteers who assisted with conducting the fieldwork, Randall Forsch (MD, MPH) and Emmet Norris, were exceptional in their helpfulness, diligence, and cleanliness. This work was funded and supported by an NSF Postdoctoral Fellowship, OCE-2126562 (KOF) and OPP-2138217 (AR), startup funds (SMA), and the John Dove Isaacs Chair in Natural Philosophy for Scripps Institution of Oceanography (SMA). The authors acknowledge that this research was conducted on land given to the National Park Service under the Alaska National Interest Lands Conservation Act of 1980 from the Chugach Alaska Corporation and earlier, from the stewardship of the Alutiiq who inhabited this coast before the glaciers pulled back from the ocean, and who maintain subsistence rights within these fjords.

## Author contributions

K.O.F. designed the study and performed the analyses. S.M.A. supervised the project and contributed to the study design. K.O.F. and S.M.A. helped fund the research. K.O.F. and A.R. collected the data. K.O.F. and S.M.A. guided the interpretation of the trace element data and analyses. KOF wrote the paper; all authors discussed the results and edited the paper.

## Competing interests

The authors declare no competing interests.
