## [Transparent Peer review file · Nature Communications]

Tidewater cycle drives alpine glacial sediment plume geochemistry

Corresponding Author: Dr Kiefer Forsch

Version 0:

Reviewer comments:

Reviewer #1

(Remarks to the Author)

This is a review of the manuscript titled "Tidewater cycle drives alpine glacial sediment plume geochemistry" by Forsch et al. for publication consideration in Nature Communications. The authors show changes in Fe and Mn lability across glacial gradients in two geomorphologically contrasting Alaskan fjord systems. The unique cycling of so-called tidewater glaciers makes for an interesting opportunity to look at the geochemical impact of advanced and retreating glaciers on sediment supply to the marine environment on timescales shorter than typical glacial retreat. To this reviewer's knowledge, this is the first study of Fe and Mn mobility/lability in these systems together in one study, with implications for the biogeochemistry of coastal Alaska.

The conclusions are generally supported by the data, if a little wandering/dense at times. I found the use of glacial/cryospheric jargon hard to parse at times without more definition. I have noted this in line comments below. Nat. Comms. is not a cryospheric journal and the audience will be broad. While it is clear the authors know a lot about these systems, to the average reader it is difficult in the introduction to follow the various definitions of glacial morphologies. It wasn't until re-reading multiple times that I understood tidewater glaciers exist as both land-terminating and marine-terminating (or in the transition between the two).

I would say that the conclusions are generally good but they don't directly address one of the fundamental imports of this work: bioavailability of Fe from glacial discharge. Lability of Fe and Mn is discussed, magnitude of flux is discussed, implications for sediment churning are discussed, but the authors neglect to comment on whether one type of glacial morphology will enhance Fe supply to this Fe-limited subarctic region. If they do, it is lost in the text. I am not really sure how trace metal ratios (mentioned in the last sentence of the paper) relate to fertilization. Those relate more to the coupling between Fe and Mn, in my mind. Are you suggesting here that an increase in Mn:Fe reflects consumption of Fe down fjord?

The methods could use some expansion, particularly around the dissolved metal analyses (lines 456-462). There are no previous methods referenced with further detail. From this paragraph alone there is no clear way to replicate these analyses. What is GSC? What are the values of GSC we should be comparing to? This is GEOTRACES jargon that not everyone will immediately understand.

I recommend this article be accepted following moderate revisions, including cleaning up text and clarifications noted in the line comments below.

Line comments:

Lines 35-36: It is clunky to use "includes" here twice, rephrase

Lines 49-51: Something about this sentence is awkwardly phrased so I don't quite follow. Do you mean that there is a large effort to quantify the flux of trace metals from glacial meltwaters? The delivery of metals alone does not logically collect quantification of endmember fluxes.

Lines 56-57: This would be a good spot to clarify to readers that advanced tidewater = marine-terminating, retreated (stable) = land-terminating. As noted above, I found this confusing as a non-glaciologist.

Line 58: “hereafter” not “herein”

Lines 65-66: Use NO₃ and PO₄ abbreviations here since they are previously defined

Lines 89-91: Inconsistent citation type (parenthetical rather than endnote)

Line 126: what does medial moraine mean?

Line 150: what is an outlet glacier?

Line 264: what does aerially exposed mean?

Line 301: I think you mean Figure S4c, not S2c

Line 365: I would add a paragraph break around here, this part is very long and hard to follow. What does proglacial mean?

Lines 373-377: Confusing as written and seems contradictory. Are you saying proglacial processes are not important or are just swamped out by other signals? What is the significance of proglacial lakes draining earlier in the year?

Line 388: As noted above, fertilization potential is mentioned a lot but not really commented on. Are higher fractions of labile Fe more like to be fertilizing? I am not clear on what the downstream geochemical effects are—it would be nice if this was summarized in the final sentence of the paragraph preceding this section.

Lines 389-391: I would rephrase this sentence, tenses are switched and using “19” vs. “six”, “three”, etc. is confusing/inconsistent.

Line 759: what do you mean x-axis is grouped by element? Are “elements” here the different size fractions/fjords? Then it seems that 2 words later element means chemical element. It also feels somewhat redundant to have both a label on the x-axis and a legend on this figure with the same information.

Line 772: I quite like this schematic and it would have been useful earlier when I was trying to sort out what marine vs land terminating meant and how that connects to tidewater glaciers. I wonder if you can add something visual about the relative magnitude of Fe and Mn supply coming from each type? For instance, it seemed like there was a lot more dFe and dMn in the NWG (retreating).

Line 786: Reference materials for the particulate phase? Lability?

Supplement comments:

Figure S5: I assume this accidentally got rotated but it is hard to read in it's current format.

Table S2: Font is very small, I would suggest splitting this up more to make it more readable for people who might print this out. As it is supplementary, you probably have the space to do this.

Reviewer #2

(Remarks to the Author)

Review of Forsch et al.

Forsch et al. conducted a geochemical analysis of suspended sediment-plume and iceberg-laden particulate matter from two distinct fjord settings in Southcentral Alaska, each representing either stable or retreating tidewater glaciers. Their findings reveal that glacial weathering processes respond differently depending on the degradation history of tidewater glaciers, with particular emphasis on the implications for future ecosystem fertilization in the context of ongoing global warming. While the results are intriguing and would contribute to the ongoing discussion on coastal ecosystem responses to cryosphere changes in polar regions, the specific aspect of this argument is not yet sufficiently developed to support the bold conclusions of the paper. Furthermore, the manuscript requires additional revisions to address numerous typographical errors and improve the overall structure.

Major concerns

#1. The elemental concentrations in this study are expressed as relative abundances to the initial mass of seawater filtered (correct? – please clarify this at the beginning). Therefore, these concentrations are largely influenced by particle concentrations and other factors such as sediment provenance, weathering regime, transport system, and post-depositional processes. The mass of particulate matter in seawater could serve as a straightforward indicator for determining terrestrial input although seawater samples collected over just two days can be significantly influenced by short-term variability, thus their particle concentrations might be not representative to interpret the overall environments in each region.

#2. Lines 150-152 : The bedrock geology is a critical factor influencing the geochemistry of the marine particulates. However, the authors dismiss this factor, reasoning that NWG and AG are outlet glaciers from the same icefield. This explanation appears insufficiently robust, as cases such as Svalbard demonstrate that even within a shared icefield, bedrock

geology can vary significantly. To address this, the authors should expand their interpretation and discussion of sediment source dynamics by incorporating evidence from previous field studies (e.g., refs. 40 and 41). Additionally, a comparison of the geochemistry of authigenic-free (i.e., detrital) fractions between the two regions would enhance their suggestion. Since detrital fractions are primarily derived from terrestrial inputs, examining their elemental ratios selected for resilience to secondary alteration could provide more compelling support for evaluating the influence of bedrock geology on the chemistry of the marine particulates.

#3. Lines 266-267 : The authors used dSi to evaluate glacial weathering intensity. However, I have significant doubts about the validity of using dSi as a proxy for glacial weathering in this context. This skepticism arises primarily because dSi concentrations tend to increase as the distance from the glacier decreases (seawater > brackish water). Given that the supply of weathering products should be greater near the source regions, lower dSi concentrations near the glacier front do not support the validity of this proxy. That is, dSi might mainly originate from seawater, as also evidenced by the generally high dSi concentrations in the Gulf of Alaska (Brown et al. 2010, *Marine Chemistry*). If this is correct, the main idea of this study should be further evaluated.

#4. Lines 273-282 : The authors argue that comminuted sediments beneath NWG experience greater chemical weathering intensity than those beneath AG for two reasons: (1) relatively higher average concentrations of dFe and dMn in NWG compared to AG, and (2) a modest correlation between dFe and dMn with dSi in NWG. However, this interpretation requires further evaluation and development. The average concentrations of dFe and dMn are not statistically distinguishable within the range of standard deviation (31.55 ± 10.65 nM dMn, 68.56 ± 61.55 nM dFe for NWG, and 23.56 ± 8.26 nM dMn, 32.77 ± 26.48 nM dFe for AG). Additionally, while dFe is probably terrestrially derived, the modest correlation does not support this interpretation. dSi cannot be reliably used to trace glacial weathering as mentioned above (major concerns #1), and dFe concentrations can also be governed by estuarine processes, such as the rapid adsorption of dFe onto particles near the mixing zone between freshwater and seawater (e.g., Boyle et al., 1977; Raiswell et al., 2018; Schroth et al., 2014; Zhang et al., 2015). The general decreasing trend in dFe concentrations with increasing distance from the glacier front at NWG may reflect the rapid removal of dFe through these estuarine processes.

#5. Lines 298-309 : The authors used the Mn/Fe ratio in the dissolved fraction to trace an excess of labile Mn in plumes and concluded that AG is more affected by anaerobic reductive dissolution processes based on the high Mn/Fe ratio. However, as shown in Table S1, the higher Mn/Fe ratio in AG is primarily due to higher Fe concentrations in NWG compared to AG, while Mn concentrations in AG are even lower than in NWG. In this context, I cannot confirm that the high Mn/Fe ratio in AG is caused by the preferential dissolution of Mn relative to Fe during anaerobic processes.

#6. Method : Chemical leaching in this study was conducted at 90°C for 2 hours. Although there is no information provided about the concentration and pH of the chemical reagent (i.e., HH solution) used for leaching, high temperature leaching can potentially cause the unintended dissolution of reactive materials (Bayon et al., 2002, *Chemical Geology* 187). The sediments in the Gulf of Alaska are likely composed of dispersed volcanic ash (Du et al., *Geochimica et Cosmochimica Acta* 193), of which dissolution during leaching may lead to biased chemical results.

Minor comments

#1. Lines 36 and 103 ... : The authors use both "tidewater glaciers" and "marine-terminating glaciers" interchangeably in the text. This might confuse readers unfamiliar with the terminology. Therefore, it is recommended to define these terms at the beginning.

#2. Line 42 : The authors state that only three fjords have active tidewater glaciers. However, Figure 1a shows four different tidewater glaciers in four different fjords. Please correct this discrepancy.

#3. Figure 1 : Some information seems to be missing. It would be helpful to add coordinate marks to the edges of the map to identify geological locations. Additionally, there is no information about the numbers in Figure 1b (sampling stations). Please define the meaning of the colors (white to blue) in the images of Figure 1b. Lastly, could you explain the reason why glaciated areas marked between Figures 1a and 1b are different (maybe different timeframe)?

#4. Line 141 : What is "this water"? It is not clear.

#5. Line 142 : There is no information about the "period" mentioned. Perhaps it refers to the sampling period? Please clarify this. Additionally, please add references for the temperature data.

#6. Lines 142, 180, 182, 184, 213 ... : There are inconsistencies in the figure citations, which make it difficult to follow the main flow of the manuscript. For example, on line 142, Figure S1 should be cited as Figure S2. These typos need to be corrected to improve clarity.

#7. Line 145 and others : The authors inconsistently use lowercase "s" (e.g., station 3 on line 145) and uppercase "S" (e.g., Station 9 on line 145) for station numbers. This inconsistency should be corrected for clarity.

#8. Lines 159-161 : This sentence might be better placed elsewhere, such as at the end of the Introduction section, or the earlier part of Results (e.g., before Line 139?).

#9. Lines 166-168 : This sentence might be better placed elsewhere.

#10. Lines 173-174 : Please include specific figure citations (e.g., Figure S3) to help readers follow the manuscript more smoothly.

#11. Lines 174-175 : Although the NO₃⁻ and PO₄³⁻ concentrations at the southernmost Station 10 are higher than those at the other stations (Lines 173-174), the authors state that "These relatively higher macronutrient concentrations persist along the northern flanks of the fjords and decrease on the southern flank"? Could it be that the term "northern flanks of the fjords" does not necessarily correspond to the northern regions of the fjords?

#12. Line 197 : The 'n' is inconsistently formatted, with some instances italicized. Ensure consistent formatting throughout the manuscript.

#13. Lines 202-204 : "The information provided is insufficient. 1) What does 'Seward, AK' refer to? If it is important, please indicate it on the map. Otherwise, provide more context in the statement. 2) Additionally, clarify which dataset—AG or NWG—is being compared to the concentrations observed during the fall along the Gulf of Alaska. 3) The AG and NWG datasets in this manuscript represent May, a time when glacier velocities are at their annual maximum. In contrast, the comparison data comes from the fall, when glacial discharge is greatest. What is the purpose of this comparison? What insights does it

provide? Please explain why these two datasets, collected during different seasons, are being compared (maybe in the discussion).

#14. Lines 206-211 : At the beginning of the paragraph, the authors refer to the elemental concentrations of the labile fraction. However, they later state, 'Overall, most particulate trace metals were refractory, with a lithogenic origin.' I find it difficult to see the connection between these two ideas. Does 'particulate trace metals' refer to trace metals in the labile fraction or to trace metals in the total particulate matter (Line 210)? If it refers to the latter, I believe the paragraph needs to be reorganized to make the ideas more straightforward and coherent."

#14. Lines 222-224 : This sentence might be better placed elsewhere, such as the Discussion section.

#15. Line 242 : Where the dataset for the bedrock samples come from?

#16. Line 247 : Is the geochemical dataset from cobbles isolated from two icebergs consistent with each other? Please compare the datasets, particularly focusing on the detrital fraction, to evaluate whether the sediment sources in both regions are similar.

#17. Lines 382-386 : The concentrations of labile materials are expressed as relative abundances to the initial mass of seawater filtered, thus it is highly affected by the mass of materials.

Version 1:

Reviewer comments:

Reviewer #1

(Remarks to the Author)

I am pleased with the revised version of the manuscript from Forsch et al., they have responded to my main concerns with sufficient detail in both the response document and the main text. They have clarified problematic jargon as well as their overall argument and relevance to fertilization/Fe biogeochemistry. Overall, I found the study well-conducted and well-reasoned and recommend it be accepted by the journal for publication.

I also read the second review and I think the authors responded well to the use of Si as a weathering tracer. Over these spatial scales and given what we know about the contrasting morphologies of the studied glaciers I think it is an acceptable tool to use here. Farther offshore in the GOA, chemical oceanographers have often assumed the high Si content in the ACC is due to glacial weathering.

One minor typo:

Line 416: "fluctuations" not "fluxuations"

Reviewer #2

(Remarks to the Author)

Review of Forsch et al.

I would like to begin by thanking the authors for their thoughtful and detailed responses to the reviewers' comments. It is clear that considerable effort was made to engage with the critiques, and several points have been clarified or improved. However, much of this effort appears to remain confined to the rebuttal letter, while the manuscript itself shows only limited development in its revised form. As a result, I did not observe any substantial or transformative changes to the manuscript. In particular, one previously raised concern remains especially pertinent: Nature Communications is not a specialist cryospheric journal, and the manuscript should therefore be accessible to a broad and interdisciplinary readership. Despite some revisions, the manuscript remains difficult to follow and would benefit from significant reorganization to improve clarity and narrative flow. In its current form, it is still challenging to interpret—even for readers familiar with the field.

(1) One major issue is the continued lack of a clear and coherent connection between the research question introduced in the introduction and the discussion presented at the end. This issue was noted in the original submission and remains insufficiently addressed. As suggested by the title, the central research question seems to be how tidewater-glacial cycles influence the chemistry of glacial sediment plumes. To clarify this, the introduction should be restructured to remove tangential content and to present a more focused explanation of what is meant by the "tidewater-glacial cycle." The discussion should then be more clearly aligned with this framing and developed with appropriate citations to guide the reader through the broader context and implications of the findings. Brief statements that simply summarize results without interpretation or reference (e.g., Lines 330–331: "NWG particles are depleted in Al with respect to crustal Ti (Figure 4a), indicating increased chemical weathering of bedrock") are unlikely to resonate with a general readership. I also strongly recommend more effective use of figures accompanied by thorough interpretation in the text.

(2) Since elemental concentrations are reported relative to the amount of seawater, most of the trends (excluding phosphate) are largely influenced by the concentration of suspended sediment and sediment provenance. If the sediment source remains constant (as the authors assumed), elemental concentrations may serve as a proxy for sediment load. While the study discusses the role of glacial weathering in controlling sediment supply, I believe that differences in fjord geomorphology—such as the contrast between narrow (NWG) and wide (AG) basins—may also play a significant role, particularly by influencing sediment residence time. For instance, although AG delivers approximately three times more meltwater than NWG (Yang et al., 2019), the relatively lower salinity observed in Northwestern Lagoon may indicate longer residence time or limited mixing in this region. These geomorphic and hydrodynamic controls should be more carefully

considered.

(3) Furthermore, as the elemental concentrations are expressed per unit seawater volume, it is difficult to distinguish chemical weathering from mechanical weathering using concentration data alone. In this context, the interpretation of greater chemical weathering beneath NWG (as discussed in Lines 278-302) appears to be largely speculative and would benefit from additional supporting evidence or more robust analytical constraints. The use of elemental ratios may provide more diagnostic information on the behavior of individual elements and their relationships to weathering intensity and processes under different tidewater-glacial conditions. While this approach is briefly mentioned, it is currently underdeveloped in the manuscript and not convincingly presented. A more rigorous and well-referenced interpretation of these proxies is necessary to substantiate the claims being made.

(4) In the revision, the authors provided new REE measurements from cobbles collected in the two fjords in an effort to address the possibility that differences in sediment chemistry could be driven by variation in source rock composition beneath each glacier. They argue that the REE patterns are largely similar between the two samples, with the exception of slight differences in Eu and Er anomalies, and interpret this as evidence that source lithology is broadly comparable. The authors further states:

“Positive Eu anomalies in granitic rocks can be associated with earlier plagioclase crystallization during magmatic differentiation (Dyger et al., 2024), perhaps signifying slight differences in the timing of plagioclase mineral crystallization in the granitic rocks beneath Aialik vs. Northwestern Glacier. Elements such as iron and manganese are considered trace elements in the mineral plagioclase (Nakada et al., 2019), and the weathering of rocks with slightly different plagioclase abundances should therefore not strongly influence the Fe and Mn flux transported to the marine environment.”

However, I find this interpretation unconvincing for three main reasons.

- a) The authors' suggestion of “differences in the timing of plagioclase crystallization” effectively implies that the source rocks beneath Aialik and Northwestern Glaciers differ in their magmatic history and mineral assemblage—hence contradicting the assertion that the source lithology is comparable.
- b) Even if the two source rocks are broadly similar in mineralogical composition aside from variations in plagioclase content, differences in the relative abundance of other mineral phases could still exert a significant influence on the mobilization of trace elements—particularly under differing chemical weathering regimes. In fact, the measured concentrations of Fe and Mn in the two cobble samples differ substantially (e.g., 1019 vs. 424 $\mu\text{mol g}^{-1}$ d.w.s. for total particulate Fe, and 33.91 vs. 18.16 $\mu\text{mol g}^{-1}$ d.w.s. for total particulate Mn), suggesting that source rock variability may still be a non-negligible factor.
- c) Most importantly, I would question the assertion that the REE patterns are largely similar between the two cobble samples. In my view, the differences are quite pronounced: the cobble from Aialik Glacier (AG) exhibits a middle rare earth element (MREE)-enriched pattern, whereas the cobble from Northwestern Glacier (NWG) shows a heavy rare earth element (HREE)-depleted pattern. These distinct signatures appear to be clearly reflected in the REE patterns of the sediments delivered from each respective source, suggesting that source rock composition may indeed influence sediment geochemistry more strongly than acknowledged in the manuscript.

(5) While most of the typographical errors highlighted previously have been corrected, a number of minor errors still persist. More critically, there are lots of inconsistencies between the chemical data presented in Supplementary Table S2 and their interpretation in the Results section. For instance, the manuscript states:

“Based on total particulate Al, a relatively immobile element found in crustal aluminosilicates, we find that NWG produces more sediment than AG as average fjord surface concentrations are nearly twice as high (NWG: $\sim 16,000$ nmol kg⁻¹, n = 10; AG: $\sim 7,900$ nmol kg⁻¹, n = 10).”

However, based on calculations from the dataset provided, the average Al concentrations appear to be approximately 7,100 nmol kg⁻¹ for NWG and 3,960 nmol kg⁻¹ for AG. This discrepancy raises concerns about the reliability of other quantitative interpretations in the manuscript, and similar inconsistencies can be found elsewhere in the data.

In summary, although the authors demonstrate a strong understanding of the system under study and have made efforts to address reviewer feedback, the manuscript continues to lack the clarity, internal consistency, and cohesion required to effectively communicate the significance of the findings. Given these persistent issues, the manuscript does not currently meet the journal's standards for publication and is not sufficiently accessible to a broad scientific audience. I therefore recommend rejection.

Minor comments

- #1. Lines 156-168: The authors consider AG to be in a stable tidewater configuration; however, its glacial flux is approximately five times greater than that of NWG, which is in a retreating configuration. This relationship is not immediately intuitive, and additional clarification would be helpful
- #2. Lines 165-166: What is the scientific significance of the peak glacier velocity observed during your sampling period?
- #3. Lines 173-175: At stations 7, 8, and 9 in NWG, NO₃ and PO₄ concentrations generally appear higher on the western flanks of the fjords, which seems to contradict your interpretation. I guess this might be influenced by glacial flux from nearby land-terminating glaciers?
- #4. Lines 190-191: This appears to contrast with your earlier statement: 'There was no observable trend in dFe concentrations with respect to distance from the glacier terminus or water column depth; (Lines 184-185).
- #5. Lines 205-207: Before this statement, I recommend clarifying the primary reason for the elevated Mn concentrations in

the inner fjord. Meanwhile, given the lack of Mn data from the inner fjord during the fall, it is difficult to draw conclusions about the potential role of a time lag in coastal exchange. As you previously noted, glacial discharge peaks during the fall, so it may be more appropriate to suggest that increased meltwater input delivers more Mn, thereby enhancing its influence in more distal areas, such as the continental shelf. It is also plausible that Mn concentrations in the inner fjord are higher during the fall.

#6. Lines 216-217, 224, 225...: not consistent with dataset presented in Supplementary Table.

#7. Lines 290-292: Intuitively, it seems that prolonged interaction between high-surface-area sediments and meltwater would most likely occur during a stable phase of the glacier. Could you provide supporting references for the statement?

#8. Lines 315-316: Please provide a clearer interpretation and relevant supporting references

#9. Line 331: Please provide a clearer interpretation and relevant supporting references

#10. Line 367: I believe that Fe is generally considered a mobile element, in contrast to Al and Ti.

#11. Line 416: fluctuations

#12. Figure 4: The authors measured elemental concentrations from two cobbles, which show substantial differences according to the Supplementary Table in my view. However, in Figure 4, the elemental ratios for the cobbles are presented as a single value, and it is unclear how this value was derived. It does not appear to be an average, nor does it correspond to data from either individual cobble. For example, the total Al/Ti and labile Al/Ti ratios for the cobbles are both shown as approximately 35, which is inconsistent with the values reported for cobble 1 (total Al/Ti = 16.85, labile Al/Ti = 33.07) and cobble 2 (total Al/Ti = 22.02, labile Al/Ti = 131.66), as well as their averages. A similar inconsistency exists for the Fe/Al (~0.3) and Mn/Fe (~0.03) ratios presented in the figure. Clarification on how these single values were calculated or derived is needed.

Version 2:

Reviewer comments:

Reviewer #2

(Remarks to the Author)

This manuscript is now in its second revision, and it is evident that the paper has improved substantially with each iteration. I particularly commend the authors for their thorough and detailed responses to the reviewers' feedback, as well as their careful incorporation of these responses into both the main text and supplementary information. At this stage, I have no further critiques regarding the data or interpretations. I would like to acknowledge the authors' diligent efforts in addressing the comments and their patience in responding to my queries. This study represents a significant contribution and is likely to stimulate further important research on tidewater glacier cycles and associated nutrient inputs to coastal oceans. I therefore recommend this manuscript for publication

Response to Editorial & Reviewer comments for NCOMMS-24-52438

Key:

Reviewer & editorial comments

Author response

We are grateful to both reviewers for their critical assessments of the research, interpretation, and writing, which has greatly improved the quality of this manuscript. Our point-by-point responses are below.

REVIEWER COMMENTS

Reviewer #1 (Remarks to the Author):

This is a review of the manuscript titled “Tidewater cycle drives alpine glacial sediment plume geochemistry” by Forsch et al. for publication consideration in Nature Communications. The authors show changes in Fe and Mn lability across glacial gradients in two geomorphologically contrasting Alaskan fjord systems. The unique cycling of so-called tidewater glaciers makes for an interesting opportunity to look at the geochemical impact of advanced and retreating glaciers on sediment supply to the marine environment on timescales shorter than typical glacial retreat. To this reviewer’s knowledge, this is the first study of Fe and Mn mobility/lability in these systems together in one study, with implications for the biogeochemistry of coastal Alaska.

The conclusions are generally supported by the data, if a little wandering/dense at times. I found the use of glacial/cryospheric jargon hard to parse at times without more definition. I have noted this in line comments below. Nat. Comms. is not a cryospheric journal and the audience will be broad. While it is clear the authors know a lot about these systems, to the average reader it is difficult in the introduction to follow the various definitions of glacial morphologies. It wasn’t until re-reading multiple times that I understood tidewater glaciers exist as both land-terminating and marine-terminating (or in the transition between the two).

I would say that the conclusions are generally good but they don’t directly address one of the fundamental imports of this work: bioavailability of Fe from glacial discharge. Lability of Fe and Mn is discussed, magnitude of flux is discussed, implications for sediment churning are discussed, but the authors neglect to comment on whether one type of glacial morphology will enhance Fe supply to this Fe-limited subarctic region. If they do, it is lost in the text. I am not really sure how trace metal ratios (mentioned in the last sentence of the paper) relate to fertilization. Those relate more to the coupling between Fe and Mn, in my mind. Are you suggesting here that an increase in Mn:Fe reflects consumption of Fe down fjord?

The methods could use some expansion, particularly around the dissolved metal analyses (lines 456-462). There are no previous methods referenced with further detail.

The methods section regarding seawater dissolved Fe and Mn concentrations (lines 546-556) have been expanded with the addition of previously published methods which outline our approach.

From this paragraph alone there is no clear way to replicate these analyses. What is GSC? What are the values of GSC we should be comparing to? This is GEOTRACES jargon that not everyone will immediately understand.

GSC has been defined and the methods referenced so that these analyses can be replicated and understood by the broader scientific community.

I recommend this article be accepted following moderate revisions, including cleaning up text and clarifications noted in the line comments below.

Line comments:

Lines 35-36: It is clunky to use “includes” here twice, rephrase

We have rephrased using “also”.

Lines 49-51: Something about this sentence is awkwardly phrased so I don't quite follow. Do you mean that there is a large effort to quantify the flux of trace metals from glacial meltwaters? The delivery of metals alone does not logically collect quantification of endmember fluxes.

While glacial meltwaters may “adversely affect microbial communities” (line 47), recognition of their importance as sources of micronutrients has now lead to global efforts to quantify the flux of trace metals and content within ice, meltwaters, and the coastal seawater.

Sentence has been changed to read: “However, the delivery of dissolved trace-metal micronutrients (iron (Fe), manganese (Mn)) to coastal and offshore marine systems from glaciated catchments has garnered large efforts to quantify the concentration of trace-metals in glacial meltwaters and fluxes to the ocean(Crusius et al., 2017).”

Lines 56-57: This would be a good spot to clarify to readers that advanced tidewater = marine-terminating, retreated (stable) = land-terminating. As noted above, I found this confusing as a non-glaciologist.

We have resolved this clarity issue and have excluded using the term ‘marine-terminating’ where we mean ‘tidewater’. They are synonymous but we used them interchangeably in the prior version.

Line 58: “hereafter” not “herein”

Grammer changed to “hereafter.”

Lines 65-66: Use NO₃ and PO₄ abbreviations here since they are previously defined

We now consistently use the abbreviations throughout the text following the definition.

Lines 89-91: Inconsistent citation type (parenthetical rather than endnote)

The citation formats were corrected.

Line 126: what does medial moraine mean?

Medial moraines are debris ridges that form when two glaciers intersect. For simplicity, “medial” was removed.

Line 150: what is an outlet glacier?

An outlet glacier is one that derives its flow from the large amounts of snow accumulation on the icefield/icecap. Under this normal stress, the ice flows through corridors carved by outlet glaciers.

Line 264: what does aerially exposed mean?

Aerially exposed sediments are those that are in contact with the atmosphere. This contrasts with sediments which are not exposed to the atmosphere, such as beneath a glacier or body of water.

We have edited this sentence to read: “However, it is not well constrained how glacial retreats impact sediment transport or the geochemistry of stored sediments; for example, tidewater glacial retreat may

result in increased transport of exposed sediment through outwash and fluvial environments.”

Line 301: I think you mean Figure S4c, not S2c

Correct. We have changed the text to reflect the correct supplemental figure.

Line 365: I would add a paragraph break around here, this part is very long and hard to follow. What does proglacial mean?

Thank you for the suggestion. We have made a paragraph break before discussing our comparisons to analogous systems. Proglacial refers to being in front of, or next to the glacier. We have added a brief definition of proglacial in this sentence.

Lines 373-377: Confusing as written and seems contradictory. Are you saying proglacial processes are not important or are just swamped out by other signals? What is the significance of proglacial lakes draining earlier in the year?

The mechanisms for mobilization of Fe are different in proglacial environments, where organic matter has a significant role in producing higher Fe-to-Al ratios. This may occur over very long timescales since this sediment is not transported. We have removed the sentence about lakes draining earlier in the year. The main point was to demonstrate a mismatch in the timing of these sources and the Spring bloom, which occurs in May, as revealed by a 14 year timeseries of coastal and off-shelf productivity (Waite and Mueter, 2013). Productivity is negatively correlated with sea surface temperature, possibly indicating an important link between cold meltwater supply and phytoplankton biomass. We have added this clarifier in the text in lines 449 to 451.

Line 388: As noted above, fertilization potential is mentioned a lot but not really commented on. Are higher fractions of labile Fe more like to be fertilizing? I am not clear on what the downstream geochemical effects are—it would be nice if this was summarized in the final sentence of the paragraph preceding this section.

Labile particulate trace metals are characterized by a higher fertilizing potential compared to total particulate trace metals, which include a large refractory component that is not solubilized under the leach conditions applied in this study. Dissolved Fe and Mn concentrations—the most bioavailable form we observed—remain two orders of magnitude higher than open-ocean concentrations observed beyond the shallowest sill. Dissolved metals are subject to processes which strongly attenuate concentrations down-fjord such as particle adsorption. Considering this pool is orders of magnitude lower than labile particulate trace metals, we limit our discussion to the particulate pool, which dominates Fe and Mn speciation.

We agree that this is an important point that we failed to sufficiently discuss. While we did not include the analysis in our discussion, the length-scales of the decay of labile particulate (Lp) trace metal (TM) concentrations (equivalent to one e -fold or decrease of 37%) can be drastically shorter than the length of the fjord (Fig. S6). This suggests that most suspended particles are removed from the water column (sedimented) within the fjord (Fig. S5). If we assume the Fe-limited region of the Gulf of Alaska is ~100 km away and the length-scale is approximately 10 km in the case of LpFe of the $>5 \mu\text{m}$ fraction in AG (Fig. S6), this implies the AG glacial plume contributes a small fraction ($\sim 0.37^{(100 \text{ km} / 10 \text{ km})}$) of the concentration of $\text{TM}_{\text{labile}}$ observed at the glacier terminus. This equates to $\sim 0.000048 * 500 \text{ nM} = 24 \text{ pM}$, which is a small fraction of the total (particulate and dissolved) Fe supplied to the offshore region from dust and the continental shelf sedimentary resuspension (Lippiatt et al., 2010). However, other efficient export mechanisms are capable of delivering fjord meltwaters to the continental shelves: during conditions when the salinity minimum occurs in October and stabilization of suspended fine particles occurs, this may buffer dissolved Fe and Mn concentrations once solubilized downstream (Milne et al., 2017). In these scenarios, fjord meltwaters would directly inject into the Alaska Coastal Current (Fig. 1).

In a study of dissolved and total dissolvable (dissolved plus labile particulates) metals, Schroth et al. (2014) demonstrated the length-scales are much shorter for the dissolved fraction compared to the total dissolvable Fe over a large salinity range (Figure 1, C and D in Schroth et al. 2014). This is direct evidence that labile particles are less prone to estuarine removal and may remain suspended within the water column for longer than the dissolved fraction, and therefore is a primary focus of this study. Biological uptake, intense scavenging, and conversion of dissolved Fe to labile particulate Fe may all explain rapid removal of dFe.

While removal of bioavailable micronutrients is efficient within these estuarine systems, we note that our sampling occurred during the springtime (May), a time of relatively little glacial meltwater input. As the glaciers continue produce meltwater throughout the onset and height of summer, concentrations of dissolved and labile particulate trace metals may increase within the fjord and scale with meltwater flux. However, our comparison of the two fjords demonstrates that retreating glaciers transport more sediments to the fjord surface with reduced chemical lability rendering them less accessible to phytoplankton and potentially abiotic solubilization (e.g., by organic ligands, photoreduction). These distinctions balance so that both fjords contain similar concentrations of labile particulate trace metals (Fig. 4d). We also note that this balance (lower lability, but higher sediment flux) may exist as a transient signal which is not sustained as transported subglacial sediments are eventually exhausted beneath NWG (retreating glacier). It would be interesting to compare concentrations at their autumnal meltwater peak. Therefore, we conclude that both fjords have similar fertilization potential in terms of labile particulate trace metals, but we anticipate with regional warming that all retreating tidewater glaciers will undergo a period of increased sediment transport, enhanced chemical weathering, and reduced chemical lability of sediments. Therefore, a key feature of the tidewater glacier cycle is that chemical lability of transported sediments traces a transient signature of the glacier degradation status.

Figure S6. Length-scales (D) of total (top) and labile (bottom) particulate trace metal concentrations. The parameter D comes from the exponential decay equation ($C_x = C_0 * e^{-x/D}$) fit to the measured concentrations (C_x) as a function of distance (x) from the glacier terminus.

We have added a summarizing sentence to the end of the preceding paragraph.

Lines 389-391: I would rephrase this sentence, tenses are switched and using “19” vs. “six”, “three”, etc. is confusing/inconsistent.

We now specify the locality of the glaciers that are adjacent to the continental shelf. There are 15 in total: four are tidewater, eight are land-terminating, and three are lake-terminating. For clarity we have excluded using marine-terminating, as this is synonymous with tidewater.

Line 759: what do you mean x-axis is grouped by element? Are “elements” here the different size fractions/fjords? Then it seems that 2 words later element means chemical element. It also feels somewhat redundant to have both a label on the x-axis and a legend on this figure with the same information.

We agree that the x-axis was redundant and have updated the figure and caption.

Line 772: I quite like this schematic and it would have been useful earlier when I was trying to sort out what marine vs land terminating meant and how that connects to tidewater glaciers. I wonder if you can add something visual about the relative magnitude of Fe and Mn supply coming from each type? For instance, it seemed like there was a lot more dFe and dMn in the NWG (retreating).

While we agree that this schematic figure is helpful, much of what we show here is gleaned from the data presented in this study and allows us to arrive at this conclusion. We instead rely on the detailed description of anticipated sediment dynamics during different stages of the tidewater glacier cycle (Lines 96 to 113) to provide more context to the reader.

Line 786: Reference materials for the particulate phase? Lability?

There are no certified or consensus values for the chemical lability of the geologic reference materials reported in this study (Table 1). However, we report the chemical lability of Fe in ATD samples to be ~11%, which is consistent with previous studies using HAc/hydroxylamine hydrochloride (12%) or oxalate/EDTA (11%) leaches (Revels et al., 2015). It is assumed, based on minerology, that both leaches access the dominant Fe oxide component of the ATD sample.

Supplement comments:

Figure S5: I assume this accidentally got rotated but it is hard to read in it's current format.

Thank you for catching this mistake. We have rotated the figure to its correct orientation.

Table S2: Font is very small, I would suggest splitting this up more to make it more readable for people who might print this out. As it is supplementary, you probably have the space to do this.

We have made the table more readable for print by splitting into two S2 tables. The tables will also be included for download in spreadsheet format.

Reviewer #2 (Remarks to the Author):

Review of Forsch et al.

Forsch et al. conducted a geochemical analysis of suspended sediment-plume and iceberg-laden particulate matter from two distinct fjord settings in Southcentral Alaska, each representing either stable or retreating tidewater glaciers. Their findings reveal that glacial weathering processes respond differently depending on the degradation history of tidewater glaciers, with particular emphasis on the implications for future ecosystem fertilization in the context of ongoing global warming. While the results are intriguing and would contribute to the ongoing discussion on coastal ecosystem responses to cryosphere changes in polar regions, the specific aspect of this argument is not yet sufficiently developed to support the bold conclusions of the paper. Furthermore, the manuscript requires additional revisions to address numerous typographical errors and improve the overall structure.

Major concerns

#1. The elemental concentrations in this study are expressed as relative abundances to the initial mass of seawater filtered (correct? – please clarify this at the beginning). Therefore, these concentrations are largely influenced by particle concentrations and other factors such as sediment provenance, weathering regime, transport system, and post-depositional processes. The mass of particulate matter in seawater could serve as a straightforward indicator for determining terrestrial input although seawater samples collected over just two days can be significantly influenced by short-term variability, thus their particle concentrations might be not representative to interpret the overall environments in each region.

While the mass of particulate matter collected on individual filters was not collected, the volume of seawater that passed through the filter was 0.25 to 0.5 L per sample, therefore the elemental concentrations reported in this study are representative of the elements transported from the glacial to the marine environment per unit volume of water. Factors that may influence the concentrations of elements in the dissolved and particulate fraction of the filtered material include: 1) the amount of sediment transported through physical weathering, 2) the chemical weathering intensity, 3) initial bedrock composition, and 4) transport system and distance from the toe of the glacier into the fjord. All these parameters (excluding initial bedrock composition) can be strongly influenced by the glacial geomorphology and gauging this influence on nutrient composition and quantity delivered to the fjord environment is a primary goal of this study. Another factor which may influence particulate metal concentrations includes post-depositional or water column removal processes (e.g., scavenging). While we acknowledge that the samples collected in this study were discrete sampling which occurred over a 2-day period, they are providing a “snapshot” in time. We agree that while longer-term sampling throughout the course of a melt season would be ideal to probe how glacial geomorphology influences nutrient transport to the marine environment, it is beyond the scope of this study. We apply metal-metal ratios and fractional labilities for the examination of trends and differences between the fjords, which are variables potentially resistant to high frequency changes in sediment transport to the fjord surface. We have acknowledged this useful comment and perspective and added text to the methods section to highlight these points.

#2. Lines 150-152 : The bedrock geology is a critical factor influencing the geochemistry of the marine particulates. However, the authors dismiss this factor, reasoning that NWG and AG are outlet glaciers from the same icefield. This explanation appears insufficiently robust, as cases such as Svalbard demonstrate that even within a shared icefield, bedrock geology can vary significantly. To address this, the authors should expand their interpretation and discussion of sediment source dynamics by incorporating evidence from previous field studies (e.g., refs. 40 and 41). Additionally, a comparison of the geochemistry of authigenic-free (i.e., detrital) fractions between the two regions would enhance their suggestion. Since detrital fractions are primarily derived from terrestrial inputs, examining their elemental ratios selected for resilience to secondary alteration could provide more compelling support for evaluating the influence of bedrock geology on the chemistry of the marine particulates.

While we were unable to directly sample the bedrock underlying both Aialik and Northwestern Glaciers, we use cobbles collected from individual fjord iceberg samples as representative of the bedrock composition, as cobbles carried by icebergs are physical transported pieces of bedrock sourced from beneath the respective glaciers. Literature indicates that the predominant bedrock beneath the Harding Icefield is Paleocene-Eocene granitic rocks, with the presence of metasedimentary rocks present on the periphery of the fjords. To investigate the variations in bedrock composition between the two fjords, we examine the rare earth element (REE) compositions of each cobble. The REE composition of sediment, dust, and bedrock are frequently used to investigate source provenance variability as the patterns of REEs fractionate as a result of differences in geochemical composition (Gabrielli et al., 2009, 2010). Shifts in REE patterns generally indicate differences in sediment or bedrock provenance, and the REE composition of cobbles from each of the fjords are generally similar (Fig. S7) with the exception of a positive Eu anomaly and a negative Er anomaly in cobbles from Aialik Glacier with respect to cobbles

from Northwestern Glacier. Positive Eu anomalies in granitic rocks can be associated with earlier plagioclase crystallization during magmatic differentiation (Dygert et al., 2024), perhaps signifying slight differences in the timing of plagioclase mineral crystallization in the granitic rocks beneath Aialik vs. Northwestern Glacier. Elements such as iron and manganese are considered trace elements in the mineral plagioclase (Nakada et al., 2019) and the weathering of rocks with slightly different plagioclase abundances should therefore not strongly influence the Fe and Mn flux transported to the marine environment. We therefore consider the subtle shifts in underlying bedrock compositions of each fjord as not a strong influencer of the resulting Fe and Mn fluxes in the marine particulates.

Figure S7. REE spider diagrams for the detrital fraction of each marine particulate sample with the color corresponding to the fjord (green = AG; orange = NWG) or cobbles (blue). Triangles represent the fine particle (<5 μm) fraction, while circles are the large particle (>5 μm) fraction.

#3. Lines 266-267 : The authors used dSi to evaluate glacial weathering intensity. However, I have significant doubts about the validity of using dSi as a proxy for glacial weathering in this context. This skepticism arises primarily because dSi concentrations tend to increase as the distance from the glacier decreases (seawater > brackish water). Given that the supply of weathering products should be greater near the source regions, lower dSi concentrations near the glacier front do not support the validity of this proxy. That is, dSi might mainly originate from seawater, as also evidenced by the generally high dSi concentrations in the Gulf of Alaska (Brown et al. 2010, Marine Chemistry). If this is correct, the main idea of this study should be further evaluated.

Previous studies have shown that the dSi flux from glacial meltwaters is large and decreases with distance from the glacial tongue (Cape et al., 2019; Kanna et al., 2018; Meire et al., 2016). We have updated the arrow in the dSi figure (Figure 2) to emphasize these same along-fjord changes in Aialik and Northwestern fjords. Within these enclosed fjord systems, we expect the predominant geochemical signature of the water and particulates to be influenced by the glacial weathering processes. We do not attribute high dSi to the presence of high-dSi Gulf of Alaska (GoA) surface waters which are located hundreds of kilometers away from the glacial fjord. We observe an offset in concentrations between the fjords, which supports that these differences are due to local glacial weathering regimes, and not water masses. This is supported by recent published work, where morphodynamics (for example, steepness) is a controlling factor in generation of dissolved solids exported through glacial streams (Jenckes et al., 2022). The publication discussed (Brown et al., 2010) does show the important influence of deep

convective mixing and upwelling of macronutrient (including dSi) enriched subsurface waters in the GoA, with a strong gradient of *decreasing* dSi towards the coast. Lippiatt et al. (2010) shows an important increase in dSi towards the coast of the Kenai Peninsula in September, indicating a coastal glacial source of dSi (Fig. 10 in Lippiatt et al., 2010). However, dSi dynamics within fjords versus continental shelves require some discussion here. New research highlights that dSi behaves non-conservatively in Greenland fjord surfaces with the presence of silicifying phytoplankton and release of amorphous silica from suspended particles, resulting in the muting of the dSi signal associated with glacier weathering (Hopwood et al., 2025). Concentrations of dSi are elevated close to glaciers in Greenland and there is also greater primary productivity in the coastal ocean of the continental shelves (by silicifying organisms) which further draw down dSi concentrations on a similar spatial scale to which it is supplied. Furthermore, such low concentrations in coastal waters of the Kenai Peninsula (~1 μM , Figure 5 in Brown et al., 2010) also highlights our point that the glaciers studied here are a source of dSi to the fjords since concentrations are 0-2.5 μM in AG and >2.5 μM in NWG. Even if GoA waters enriched in dSi can enter the fjords at depth (not easily considering the shallow fjord sill depths), only subsurface water in AG would be upwelled due to positively buoyant meltwater injection at the glacier face. Such mechanism is absent in NWG, which inputs meltwater directly to the surface. Furthermore, if the elevated dSi signal was from GoA waters, we would also observe correlation with other macronutrients, phosphate and nitrate, which we did not observe (Fig. S3). In the absence of differences in productivity within the fjords studied here (discussed lines 373 to 383), and without a large component of GoA surface waters entering the fjords, we attribute dSi concentration differences to weathering regimes.

#4. Lines 273-282 : The authors argue that comminuted sediments beneath NWG experience greater chemical weathering intensity than those beneath AG for two reasons: (1) relatively higher average concentrations of dFe and dMn in NWG compared to AG, and (2) a modest correlation between dFe and dMn with dSi in NWG. However, this interpretation requires further evaluation and development. The average concentrations of dFe and dMn are not statistically distinguishable within the range of standard deviation (31.55 ± 10.65 nM dMn, 68.56 ± 61.55 nM dFe for NWG, and 23.56 ± 8.26 nM dMn, 32.77 ± 26.48 nM dFe for AG). Additionally, while dFe is probably terrestrially derived, the modest correlation does not support this interpretation. dSi cannot be reliably used to trace glacial weathering as mentioned above (major concerns #1), and dFe concentrations can also be governed by estuarine processes, such as the rapid adsorption of dFe onto particles near the mixing zone between freshwater and seawater (e.g., Boyle et al., 1977; Raiswell et al., 2018; Schroth et al., 2014; Zhang et al., 2015). The general decreasing trend in dFe concentrations with increasing distance from the glacier front at NWG may reflect the rapid removal of dFe through these estuarine processes.

If we assume that detrital particulate Al is a good tracer for total suspended sediment mass in the marine environment (e.g., Ohnemus and Lam, 2015), then NWG produces a sediment plume with a higher suspended mass concentration. As this reviewer comment pointed out, this would provide ample opportunity to scavenge particle-reactive elements, such as dFe and dMn. Higher concentrations of dFe in NWG are subject to the reactive surface area of particles, yet these high concentrations are maintained at the surface. The same follows for dMn, however, dMn has a longer residence time in the surface ocean due to photochemical production of Mn(II) (Sunda and Huntsman, 1994), and therefore may be resistant to such intense scavenging processes. For this reason, dMn may be a better indicator of recent dFe input from anoxic sources (Sherrell et al., 2018), for which there are on-average greater dFe concentrations at NWG than in AG, especially at stations closest to the glacier terminus (Stations 1 – 3). The correspondingly greater ratios of dissolved Mn/Fe compared to labile particulate Mn/Fe in both fjords demonstrate the process of rapid removal of dFe through formation of authigenic minerals and scavenging onto particles.

#5. Lines 298-309 : The authors used the Mn/Fe ratio in the dissolved fraction to trace an excess of labile Mn in plumes and concluded that AG is more affected by anaerobic reductive dissolution processes based on the high Mn/Fe ratio. However, as shown in Table S1, the higher Mn/Fe ratio in AG is primarily due to higher Fe concentrations in NWG compared to AG, while Mn concentrations in AG are even lower than in NWG. In this

context, I cannot confirm that the high Mn/Fe ratio in AG is caused by the preferential dissolution of Mn relative to Fe during anaerobic processes.

Thank you for this comment, which we reflected upon and determined that we lack sufficient data to distinguish whether the source of dissolved Fe and Mn is anoxic. For this reason, we have decided to restrict our interpretation to the labile particles.

#6. Method : Chemical leaching in this study was conducted at 90°C for 2 hours. Although there is no information provided about the concentration and pH of the chemical reagent (i.e., HH solution) used for leaching, high temperature leaching can potentially cause the unintended dissolution of reactive materials (Bayon et al., 2002, *Chemical Geology* 187). The sediments in the Gulf of Alaska are likely composed of dispersed volcanic ash (Du et al., *Geochimica et Cosmochimica Acta* 193), of which dissolution during leaching may lead to biased chemical results.

As described in Berger et al., (2008) with modifications made by Milne et al., (2017), the chemical leach was conducted over two hours, with an initial 30 minute heating step at 90°C. We have included more details about the chemical leach in the text in line 568 to 569. Glacial ash is a potentially important yet temporally inconsistent contributor of trace metals to the marine environment in the Gulf of Alaska, as volcanic eruptions are relatively sporadic events rather than constant contributors of trace metals through physical erosion processes such as glacial grinding of bedrock. For example, Mount Spurr, a stratovolcano located in Southcentral Alaska last erupted in 1992 and may erupt again imminently (<https://avo.alaska.edu/volcano/spurr>) resulting in potentially two volcanic ash depositing events in the past 3 decades. While we acknowledge that volcanic ash would be a strong contributor of metals to the surface ocean, much of the sediments supplied to the fjords which feed into the Gulf of Alaska are glaciogenic and to a lesser degree biogenic in origin, with discrete volcanic events indistinguishable from the glaciogenic sediment in the sense that the geochemical composition of the volcanic ash and glaciogenic sediment may be similar in composition (Davies et al., 2011). The sediments that we analyze in this study are suspended marine particles in the glacial fjords and are predominately glaciogenic in origin based on the proximity to the glacial terminus and the absence of volcanic eruptions during the sampling time interval.

Minor comments

#1. Lines 36 and 103 ...: The authors use both "tidewater glaciers" and "marine-terminating glaciers" interchangeably in the text. This might confuse readers unfamiliar with the terminology. Therefore, it is recommended to define these terms at the beginning.

We have decided to use only “tidewater glacier” terminology to not confuse the reader.

#2. Line 42 : The authors state that only three fjords have active tidewater glaciers. However, Figure 1a shows four different tidewater glaciers in four different fjords. Please correct this discrepancy.

We note that the Holgate arm (of Aialik fjord) has a prominent active tidewater glacier (Holgate Glacier), that is contained within the geographical location of Aialik fjord.

#3. Figure 1 : Some information seems to be missing. It would be helpful to add coordinate marks to the edges of the map to identify geological locations. Additionally, there is no information about the numbers in Figure 1b (sampling stations). Please define the meaning of the colors (white to blue) in the images of Figure 1b. Lastly, could you explain the reason why glaciated areas marked between Figures 1a and 1b are different (maybe different timeframe)?

We have updated the figure to include coordinate marks to the edges of the station map and have included in the caption that the numerical values are sampling stations. We removed the gradient chronometers from Figure 1b. The differences between the areas are that in 1a, blue corresponds to

glaciers, while in 1b, the white areas are from LandSat image and show areas with snow cover in addition to glaciers at the time of download and field work.

#4. Line 141 : What is “this water”? It is not clear.

“This water” refers to seawater, which might remain in the inner fjord for long periods before being flushed seaward of the shallowest sill. We have changed to “seawater.”

#5. Line 142 : There is no information about the “period” mentioned. Perhaps it refers to the sampling period? Please clarify this. Additionally, please add references for the temperature data.

“Period” refers to the “sampling period.” We have changed the text to reflect this. Sea surface temperature data were collected as part of the field sampling.

#6. Lines 142, 180, 182, 184, 213 ... : There are inconsistencies in the figure citations, which make it difficult to follow the main flow of the manuscript. For example, on line 142, Figure S1 should be cited as Figure S2. These typos need to be corrected to improve clarity.

We have rearranged the figures in the supplemental to match the text and have fixed any discrepancies with reference to figures.

#7. Line 145 and others : The authors inconsistently use lowercase “s” (e.g., station 3 on line 145) and uppercase “S” (e.g., Station 9 on line 145) for station numbers. This inconsistency should be corrected for clarity.

We now have made all station references as lower-case.

#8. Lines 159-161 : This sentence might be better placed elsewhere, such as at the end of the Introduction section, or the earlier part of Results (e.g., before Line 139?).

We agree and have moved this sentence to an earlier paragraph, now lines 154 – 156.

#9. Lines 166-168 : This sentence might be better placed elsewhere.

We have relocated this sentence to the end of the first Results paragraph.

#10. Lines 173-174 : Please include specific figure citations (e.g., Figure S3) to help readers follow the manuscript more smoothly.

We have added a citation to the corresponding supplemental figure (S3).

#11. Lines 174-175 : Although the NO₃⁻ and PO₄³⁻ concentrations at the southernmost Station 10 are higher than those at the other stations (Lines 173-174), the authors state that ‘These relatively higher macronutrient concentrations persist along the northern flanks of the fjords and decrease on the southern flank’? Could it be that the term ‘northern flanks of the fjords’ does not necessarily correspond to the northern regions of the fjords?

We have decided it is better to indicate these flanks as “eastern flanks” since the along-fjord axes are oriented in the North/South and Northwest/Southeast directions. This would mean the flanks to which we refer are on the eastern boundary. We have made sure this is consistent in the rest of the Results section.

#12. Line 197 : The ‘n’ is inconsistently formatted, with some instances italicized. Ensure consistent formatting throughout the manuscript.

We have ensured consistent formatting for sample set size as italicized “n”.

#13. Lines 202-204 : "The information provided is insufficient. 1) What does 'Seward, AK' refer to? If it is important, please indicate it on the map. Otherwise, provide more context in the statement. 2) Additionally, clarify which dataset—AG or NWG—is being compared to the concentrations observed during the fall along the Gulf of Alaska. 3) The AG and NWG datasets in this manuscript represent May, a time when glacier velocities are at their annual maximum. In contrast, the comparison data comes from the fall, when glacial discharge is greatest. What is the purpose of this comparison? What insights does it provide? Please explain why these two datasets, collected during different seasons, are being compared (maybe in the discussion).

Thank you for these points and we hope that we have addressed these comments with clarity in the following ways:

- 1) **We referenced Seward, AK as a geographical point to provide context for a historical hydrographic timeseries, which records, among many variables, ocean salinity (proxy for meltwater and precipitation). For clarity, we remove any mention to not confuse the reader and remain consistent with the main message, which is as follows:**
- 2) **Both AG and NWG have similar concentrations of dissolved Fe and Mn to Gulf of Alaska surface waters in the fall.**
- 3) **This comparison highlights the importance of advective processes, which transport glacial meltwaters produced in the Spring, to the Gulf of Alaska continental shelf. These processes of coastal water exchange are critically important for the transport of meltwater dissolved and particulate trace elements derived from the glacial weathering of continental crust.**

We have made these changes and emphasized our point in the new version of the discussion.

#14. Lines 206-211 : At the beginning of the paragraph, the authors refer to the elemental concentrations of the labile fraction. However, they later state, 'Overall, most particulate trace metals were refractory, with a lithogenic origin.' I find it difficult to see the connection between these two ideas. Does 'particulate trace metals' refer to trace metals in the labile fraction or to trace metals in the total particulate matter (Line 210)? If it refers to the latter, I believe the paragraph needs to be reorganized to make the ideas more straightforward and coherent.

To be more coherent, we have reorganized the section “*Suspended particulate trace-metals*” (Lines 243-260) to first discuss total particulate trace-metals, and then the leached fraction.

#14. Lines 222-224 : This sentence might be better placed elsewhere, such as the Discussion section.

We have moved this sentence to the Discussion section (lines 404-406).

#15. Line 242 : Where does the dataset for the bedrock samples come from?

Individual cobbles were collected from two icebergs, one from each fjord. These cobbles represent the “rock weathered by outlet glaciers” (line 300). The dataset is original to this research.

#16. Line 247 : Is the geochemical dataset from cobbles isolated from two icebergs consistent with each other? Please compare the datasets, particularly focusing on the detrital fraction, to evaluate whether the sediment sources in both regions are similar.

We have included an extended discussion of the geochemical dataset from the analysis of cobbles above. In brief, the differences in the refractory (detrital) component of cobbles are minor with respect to the elements of interest (Al, Ti, Fe, Mn), but small variations in REEs are discussed.

#17. Lines 382-386 : The concentrations of labile materials are expressed as relative abundances to the initial mass of seawater filtered, thus it is highly affected by the mass of materials.

The filtered particles were standardized based on the volume of seawater filtered, therefore the elemental concentrations reported in this study are representative of glacially derived metals transported to the marine environment per unit volume of water. Factors that may influence the concentrations of elements in the particulate phase include: 1) the amount of sediment transported through physical weathering; 2) the subglacial chemical weathering intensity; 3) initial bedrock composition; 4) transport system and distance from glacial terminus to fjord environment. All these parameters excluding bedrock composition can be strongly influenced by glacial setting and geomorphology, and gauging this influence is a primary goal of this study. It is important to note that the samples presented in this study represent a snapshot in time and are likely to fluctuate over the course of the melt season.

References

- Berger, C. J. M., Lippiatt, S. M., Lawrence, M. G. and Bruland, K. W.: Application of a chemical leach technique for estimating labile particulate aluminum, iron, and manganese in the Columbia River plume and coastal waters off Oregon and Washington, *J. Geophys. Res. Ocean.*, 113(C2), doi:<https://doi.org/10.1029/2007JC004703>, 2008.
- Brown, M. T., Lippiatt, S. M. and Bruland, K. W.: Dissolved aluminum, particulate aluminum, and silicic acid in northern Gulf of Alaska coastal waters: Glacial/riverine inputs and extreme reactivity, *Mar. Chem.*, 122(1), 160–175, doi:<https://doi.org/10.1016/j.marchem.2010.04.002>, 2010.
- Cape, M. R., Vernet, M., Pettit, E. C., Wellner, J., Truffer, M., Akie, G., Domack, E., Leventer, A., Smith, C. R. and Huber, B. A.: Circumpolar Deep Water Impacts Glacial Meltwater Export and Coastal Biogeochemical Cycling Along the West Antarctic Peninsula, *Front. Mar. Sci.*, 6, 144, 2019.
- Crusius, J., Schroth, A. W., Resing, J. A., Cullen, J. and Campbell, R. W.: Seasonal and spatial variabilities in northern Gulf of Alaska surface water iron concentrations driven by shelf sediment resuspension, glacial meltwater, a Yakutat eddy, and dust, *Global Biogeochem. Cycles*, 31(6), 942–960, doi:<https://doi.org/10.1002/2016GB005493>, 2017.
- Davies, M. H., Mix, A. C., Stoner, J. S., Addison, J. A., Jaeger, J., Finney, B. and Wiest, J.: The deglacial transition on the southeastern Alaska Margin: Meltwater input, sea level rise, marine productivity, and sedimentary anoxia, *Paleoceanography*, 26(2), doi:<https://doi.org/10.1029/2010PA002051>, 2011.
- Dygert, N., Ustunisik, G. K. and Nielsen, R. L.: Europium in plagioclase-hosted melt inclusions reveals mantle melting modulates oxygen fugacity, *Nat. Commun.*, 15(1), 3033, doi:[10.1038/s41467-024-47224-5](https://doi.org/10.1038/s41467-024-47224-5), 2024.
- Gabrielli, P., Planchon, F., Barbante, C., Boutron, C. F., Petit, J. R., Bulat, S., Hong, S., Cozzi, G. and Cescon, P.: Ultra-low rare earth element content in accreted ice from sub-glacial Lake Vostok, Antarctica, *Geochim. Cosmochim. Acta*, 73(20), 5959–5974, doi:<https://doi.org/10.1016/j.gca.2009.05.050>, 2009.
- Gabrielli, P., Wegner, A., Petit, J. R., Delmonte, B., De Deckker, P., Gaspari, V., Fischer, H., Ruth, U., Kriews, M., Boutron, C., Cescon, P. and Barbante, C.: A major glacial-interglacial change in aeolian dust composition inferred from Rare Earth Elements in Antarctic ice, *Quat. Sci. Rev.*, 29(1), 265–273, doi:<https://doi.org/10.1016/j.quascirev.2009.09.002>, 2010.
- Hopwood, M. J., Carroll, D., Gu, Y., Huang, X., Krause, J., Cozzi, S., Cantoni, C., Gastelu Barcena, M. F., Carroll,

- S. and Körtzinger, A.: A Close Look at Dissolved Silica Dynamics in Disko Bay, West Greenland, *Global Biogeochem. Cycles*, 39(1), e2023GB008080, doi:<https://doi.org/10.1029/2023GB008080>, 2025.
- Jenckes, J., Ibarra, D. E. and Munk, L. A.: Concentration-Discharge Patterns Across the Gulf of Alaska Reveal Geomorphological and Glacierization Controls on Stream Water Solute Generation and Export, *Geophys. Res. Lett.*, 49(1), e2021GL095152, doi:<https://doi.org/10.1029/2021GL095152>, 2022.
- Kanna, N., Sugiyama, S., Ohashi, Y., Sakakibara, D., Fukamachi, Y. and Nomura, D.: Upwelling of Macronutrients and Dissolved Inorganic Carbon by a Subglacial Freshwater Driven Plume in Bowdoin Fjord, Northwestern Greenland, *J. Geophys. Res. Biogeosciences*, 123(5), 1666–1682, doi:<https://doi.org/10.1029/2017JG004248>, 2018.
- Lippiatt, S. M., Lohan, M. C. and Bruland, K. W.: The distribution of reactive iron in northern Gulf of Alaska coastal waters, *Mar. Chem.*, doi:10.1016/j.marchem.2010.04.007, 2010.
- Meire, L., Meire, P., Struyf, E., Krawczyk, D. W., Arendt, K. E., Yde, J. C., Juul Pedersen, T., Hopwood, M. J., Rysgaard, S. and Meysman, F. J. R.: High export of dissolved silica from the Greenland Ice Sheet, *Geophys. Res. Lett.*, 43(17), 9173–9182, doi:<https://doi.org/10.1002/2016GL070191>, 2016.
- Milne, A., Schlosser, C., Wake, B. D., Achterberg, E. P., Chance, R., Baker, A. R., Forryan, A. and Lohan, M. C.: Particulate phases are key in controlling dissolved iron concentrations in the (sub)tropical North Atlantic, *Geophys. Res. Lett.*, 44(5), 2377–2387, doi:<https://doi.org/10.1002/2016GL072314>, 2017.
- Nakada, R., Sato, M., Ushioda, M., Tamura, Y. and Yamamoto, S.: Variation of Iron Species in Plagioclase Crystals by X-ray Absorption Fine Structure Analysis, *Geochemistry, Geophys. Geosystems*, 20(11), 5319–5333, doi:<https://doi.org/10.1029/2018GC008131>, 2019.
- Ohnemus, D. C. and Lam, P. J.: Cycling of lithogenic marine particles in the US GEOTRACES North Atlantic transect, *Deep Sea Res. Part II Top. Stud. Oceanogr.*, 116, 283–302, doi:<https://doi.org/10.1016/j.dsr2.2014.11.019>, 2015.
- Sherrell, R. M., Annett, A. L., Fitzsimmons, J. N., Rocanova, V. J. and Meredith, M. P.: A “shallow bathtub ring” of local sedimentary iron input maintains the Palmer Deep biological hotspot on the West Antarctic Peninsula shelf, *Philos. Trans. R. Soc. A Math. Phys. Eng. Sci.*, doi:10.1098/rsta.2017.0171, 2018.
- Sunda, W. G. and Huntsman, S. A.: Photoreduction of manganese oxides in seawater, *Mar. Chem.*, 46(1), 133–152, doi:[https://doi.org/10.1016/0304-4203\(94\)90051-5](https://doi.org/10.1016/0304-4203(94)90051-5), 1994.
- Waite, J. N. and Mueter, F. J.: Spatial and temporal variability of chlorophyll-a concentrations in the coastal Gulf of Alaska, 1998–2011, using cloud-free reconstructions of SeaWiFS and MODIS-Aqua data, *Prog. Oceanogr.*, 116, 179–192, doi:<https://doi.org/10.1016/j.pocean.2013.07.006>, 2013.

Response to Reviewer Comments

We would like to thank both reviewers for their effort to improve the interpretation of the data and impact of its central message, through rigorous commentary on its components. As with the last round of reviews, the reviewer comment responses are written in **bold font** following each comment.

Begin Reviewer #2 comments:

Review of Forsch et al.

I would like to begin by thanking the authors for their thoughtful and detailed responses to the reviewers' comments. It is clear that considerable effort was made to engage with the critiques, and several points have been clarified or improved. However, much of this effort appears to remain confined to the rebuttal letter, while the manuscript itself shows only limited development in its revised form. As a result, I did not observe any substantial or transformative changes to the manuscript. In particular, one previously raised concern remains especially pertinent: *Nature Communications* is not a specialist cryospheric journal, and the manuscript should therefore be accessible to a broad and interdisciplinary readership. Despite some revisions, the manuscript remains difficult to follow and would benefit from significant reorganization to improve clarity and narrative flow. In its current form, it is still challenging to interpret—even for readers familiar with the field.

- (1) One major issue is the continued lack of a clear and coherent connection between the research question introduced in the introduction and the discussion presented at the end. This issue was noted in the original submission and remains insufficiently addressed. As suggested by the title, the central research question seems to be how tidewater-glacial cycles influence the chemistry of glacial sediment plumes. To clarify this, the introduction should be restructured to remove tangential content and to present a more focused explanation of what is meant by the “tidewater-glacial cycle.” The discussion should then be more clearly aligned with this framing and developed with appropriate citations to guide the reader through the broader context and implications of the findings. Brief statements that simply summarize results without interpretation or reference (e.g., Lines 330–331: “NWG particles are depleted in Al with respect to crustal Ti (Figure 4a), indicating increased chemical weathering of bedrock”) are unlikely to resonate with a general readership. I also strongly recommend more effective use of figures accompanied by thorough interpretation in the text.

Thank you for these constructive comments, they have improved the quality of the manuscript and allowed us to make a more comprehensive connection between the research question brought up in the introduction and what is covered throughout the discussion. To address the major issue of connecting our question about the impacts of glacial setting and TGC on coastal biogeochemistry, we have included this additional interpretation and references to the text in the section entitled, “Glacial setting drives downstream sediment geochemistry”. Specifically, the added text summarizes the geochemistry of each fjord environment and how this relates to and is driven by glacial geomorphology. (Lines 334-352)

To address the point regarding brief statements which summarize results without interpretation or reference, we have taken this feedback into strong consideration and revised large portions of the manuscript to provide additional context to guide the reader through the data and subsequent interpretation. For example, regarding the question of the relative impacts of chemical weathering and transport on the geochemistry of suspended sediments, we apply the Al/Ti ratio and explain the use of it in previous studies and how it aids in the interpretation of the data presented here. The Al/Ti ratio of sediments is a useful indicator of particle sorting, where coarse particles enriched in primary minerals such as Fe-Ti oxides (and characterized by low Al/Ti) are preferentially lost during transport as a result of density driven sorting (e.g., Aarons et al., 2023; Bouchez et al., 2011). The labile particulate Al/Ti ratio is a useful indicator of the degree of chemical weathering, whereby

Al-oxides are the ultimate product of intense chemical weathering of aluminosilicates (high labile particulate Al/Ti, Kryc et al., 2003). For context, we report the expected sediment sources supplied to each glacier sediment plume, detrital Al/Ti ratios in fjord surface sediments, the percent contribution of fine particles to the total suspended particulate concentration, and the labile particulate Al/Ti as an indicator of Al-oxides content. We describe in detail the trends in the Al/Ti ratios for the individual fjords and provide more interpretation in Lines 334-352 for how this is driven by glacial geomorphology. We summarize the details below (included in the Supplementary Material discussion “Impacts of subglacial mechanical and chemical weathering on Al-to-Ti ratios”):

In AG, the presence of a subsurface buoyant meltwater plume should result in highly sorted subglacial sediments with a high contribution to the surface sediment plume. We also expect that through vigorous mixing with surrounding seawater, the rising buoyant plume would resuspend and entrain marine sediments. Finally, melting icebergs would contribute directly to the surface signature, although concentrations are too small to explain the high concentrations observed at the fjord surface. We find AG refractory (detrital) Al/Ti of sediments to be on-average 28 ± 8 and 34 ± 8 mol:mol for small and large particle fractions, respectively. Small particles contribute on-average 50% and 53% of the total particulate concentrations of Al and Ti, respectively. The labile particulate Al/Ti ratios are on-average 24 ± 21 and 18 ± 8 for the small and large particle fractions, respectively.

In NWG, subglacial meltwaters discharge directly to the fjord surface, with no opportunity for fluvial sorting to occur. Icebergs would contribute to the fjord surface metal concentrations, but likely only a minor contribution as the sediment plume contains orders of magnitude greater sediment fluxes. We find NWG refractory (detrital) Al/Ti of sediments to be on-average 21 ± 0.6 and 24 ± 1 mol:mol for small ($< 5 \mu\text{m}$) and large ($> 5 \mu\text{m}$) particle fractions, respectively. Small particles contribute on-average 44% and 47% of the total particulate concentrations of Al and Ti, respectively. The labile particulate Al/Ti ratios are on-average 38 ± 21 and 28 ± 11 for the small and large particle fractions, respectively.

The purpose of interrogating the Al/Ti ratio in the refractory and labile particulate fractions is to understand the intensity of subglacial weathering and transport processes at different stages of the TGC. The higher Al/Ti ratios in detrital particles from AG indicates a higher degree of particle sorting, where coarse grains (characterized by enrichment in Ti) are lost from suspension (settled/sedimented) during transport from the subglacial environment to the fjord surface. The higher degree of particle sorting is also reflected in a larger contribution of small particles to the total suspended sediments in AG, compared to NWG. Finally, as Al-oxides are encompassed within the chemical extraction for labile particulate trace metals, we observed NWG particles to contain a greater contribution of Al-oxides compared to AG sediments, indicating that both particle size fractions have been extensively chemically weathered, albeit large variability exists between stations. A discussion on chemical weathering of subglacial sediments is included in response to Reviewer 2 comment #3.

- (2) Since elemental concentrations are reported relative to the amount of seawater, most of the trends (excluding phosphate) are largely influenced by the concentration of suspended sediment and sediment provenance. If the sediment source remains constant (as the authors assumed), elemental concentrations may serve as a proxy for sediment load. While the study discusses the role of glacial weathering in controlling sediment supply, I believe that differences in fjord geomorphology—such as the contrast between narrow (NWG) and wide (AG) basins—may also play a significant role, particularly by influencing sediment residence time. For instance, although AG delivers approximately three times more meltwater than NWG (Yang et al., 2019), the relatively lower salinity observed in Northwestern Lagoon may indicate longer residence time or limited mixing in this region. These geomorphic and hydrodynamic controls should be more carefully considered.

Thank you for this comment. We acknowledge that hydrodynamics may play an important role, for

example, through restricting lateral advection and residence time of meltwaters in the inner fjord. Since we lack a detailed physical description of these fjords, these details remain somewhat speculative. However, we review what is known about fjord circulation and place this within the context of the physical setting of Aialik and Northwestern fjords. Wide fjords may form eddies, which allow recirculation and retention of surface waters (Zhao et al., 2023). It is currently not clear if either fjord allows for eddy formation and retention of sediment plumes. However, based on the results of Zhao et al. (2023), NWG is extremely narrow (~1 km width) which should prevent formation of currents which recirculate meltwaters within the inner fjord. Therefore, the suggestion that higher concentrations of total particulate metals within NWG is due to a higher residence time of meltwaters remains unsupported. The lower salinity in NWG could be due to direct input of meltwaters to the surface, whereas meltwater injection in AG occurs below the surface, and must mix and entrain additional seawater as it rises to the surface. Presently, we do not know the extent of seawater entrainment from turbulent mixing in the AG buoyant meltwater plume.

We have included the following sentence within the text when discussing differences in the particulate concentrations.

“While hydrodynamics may play a role in the retention of sediments within the inner fjords through restricting lateral advection, recirculation is not expected to occur in narrow fjords (Zhao et al., 2023) and therefore cannot explain the lower salinities observed and higher total particulate trace metals in NWG.” (Lines 375-378)

- (3) Furthermore, as the elemental concentrations are expressed per unit seawater volume, it is difficult to distinguish chemical weathering from mechanical weathering using concentration data alone. In this context, the interpretation of greater chemical weathering beneath NWG (as discussed in Lines 278- 302) appears to be largely speculative and would benefit from additional supporting evidence or more robust analytical constraints. The use of elemental ratios may provide more diagnostic information on the behavior of individual elements and their relationships to weathering intensity and processes under different tidewater-glacial conditions. While this approach is briefly mentioned, it is currently underdeveloped in the manuscript and not convincingly presented. A more rigorous and well- referenced interpretation of these proxies is necessary to substantiate the claims being made.

We agree that the use of elemental ratios provides diagnostic information on the behavior during subglacial weathering and transport processes, including deposition within the fjord. Given that elemental ratios have been applied to understand the physical and chemical weathering processes at play at other coastal polar systems, we apply them similarly. The following lines of evidence are presented in support of our interpretation for greater chemical weathering and sediment transport in NWG, which are distinct from indicators of mechanical weathering:

- i) **A comparison of the La/Sm ratios of the refractory components of sediments is useful for determining the extent of chemical weathering (Wei et al., 2006). We find that sediments in NWG have on-average higher values of La/Sm (Figure S9), indicating more extensive chemical weathering processes occurring within this fjord system, as heavy REEs are depleted in the products after extreme weathering relative to light REEs (Nesbitt, 1979; Wei et al., 2006). It is important to note that the La/Sm ratio in cobbles are similar between the fjords (Figure S8), indicating La/Sm ratios of sediments result from modification of a similar bedrock source.**

Figure S9. The La-to-Sm ratios (mol:mol), an indicator of the intensity of chemical weathering (Wei et al., 2006) is plotted for all stations. Horizontal lines represent the average La-to-Sm (mol:mol) for each particle size class.

ii) Despite differences in the chemical composition of the cobbles sampled from each fjord, we see on-average elevated Fe chemical lability (expressed as a fraction of the total particulate Fe), indicating that glacier weathering is the dominant control on Fe lability. This interpretation is supported by the findings of a previous study which examined the dFe concentration in meltwater streams draining the Greenland ice sheet, across a large range of lithologies (Aciego et al., 2015). The conclusion of this study demonstrated that water-rock interaction time (residence time of sediments) in the subglacial environment is the dominant control of the concentrations of dissolved metals, rather than underlying bedrock geochemistry.

Although we do not have firm constraints on the underlying bedrock geochemistry beneath Aialik and NWG, we assume based on available literature that granite is the predominant lithology (Dumoulin, 1987; Tysdal, R.G.; Case, 1979). The accessory minerals present within granitic rock are the most likely to chemically weather first, which would result in the preferential loss of these minerals during chemical weathering. The order of crystallization of granitic rocks follows: accessory minerals, ferromagnesian minerals, lime-alkali feldspar, alkali feldspar, and finally quartz (Naney and Swanson, 1980), and the order of chemical weathering should be the opposite trend. The addition of Fe during magmatic crystallization is attributable to the rapid formation of phyllosilicates, which for the Harding Icefield includes the dominant mineral, biotite. Chemical weathering of biotite is the likely source for the high dissolved Fe and Si concentrations observed in other subglacial systems (Pryer et al., 2020; Schroth et al., 2011), which has been documented to chemically weather and be transported over periods of 145-275 years in proglacial environments and during early stages of glacial retreat (Föllmi et al., 2009). Only in the oldest sediments does silicate weathering dominate (once carbonates have been removed and requiring prolonged periods of water-rock interaction), whereas biotite weathering occurs rapidly in the subglacial environment (Anderson et al., 2000). Aluminum oxides are considered the ultimate product of chemical weathering of these aluminosilicates, and we show that NWG has on-average greater Al chemical lability, compared to AG. This chemical lability is attributed to the presence of Al-oxides in addition to possible scavenging of dAl onto Fe- and Mn-oxides. In both fjords, Al is enriched in the labile particulate phase, relative to Ti, indicating a greater adsorbed fraction of Al as Al-oxides relative to Ti (Kryc et al., 2003). The degree to which Al is enriched in particulate matter as Al-oxides is greatest in NWG, indicating intense chemical weathering has occurred or is occurring.

iii) Hydrodynamic sorting may also contribute to the observed trends, with certain elements associated with the heavy/light and coarse/fine phases. In recent investigations, the Al/Ti ratio is applied to investigate the loss of heavy minerals associated with preferential gravitational settling of coarse-sized grains which contain Ti (e.g., rutile) due to hydrodynamic sorting (e.g., Aarons et al., 2023; Bouchez et al., 2011). The refractory (detrital) fraction of the sediments within AG has greater

Al/Ti ratios compared to those in NWG. Combined with a greater contribution of the small particle size fraction to the overall concentrations of both Al and Ti, we infer that a higher degree of hydrodynamic sorting has occurred on AG suspended sediments. This is unsurprising since injection of sediment-laden subglacial sediments occurs in the subsurface, escapes intense sedimentation processes at the grounding line, and must rise to the fjord surface because of its freshwater composition and relatively lower density than seawater (Eidam et al., 2019; Syvitski, 1989).

We have added discussion regarding this point to the Supplementary text under the heading “On the diagnosis of subglacial weathering and transport processes.”

- j) In the revision, the authors provided new REE measurements from cobbles collected in the two fjords in an effort to address the possibility that differences in sediment chemistry could be driven by variation in source rock composition beneath each glacier. They argue that the REE patterns are largely similar between the two samples, with the exception of slight differences in Eu and Er anomalies, and interpret this as evidence that source lithology is broadly comparable. The authors further states:

“Positive Eu anomalies in granitic rocks can be associated with earlier plagioclase crystallization during magmatic differentiation (Dygert et al., 2024), perhaps signifying slight differences in the timing of plagioclase mineral crystallization in the granitic rocks beneath Aialik vs. Northwestern Glacier. Elements such as iron and manganese are considered trace elements in the mineral

plagioclase (Nakada et al., 2019), and the weathering of rocks with slightly different plagioclase abundances should therefore not strongly influence the Fe and Mn flux transported to the marine environment.”.

However, I find this interpretation unconvincing for three main reasons.

- a. The authors' suggestion of "differences in the timing of plagioclase crystallization" effectively implies that the source rocks beneath Aialik and Northwestern Glaciers differ in their magmatic history and mineral assemblage—hence contradicting the assertion that the source lithology is comparable.

We brought up the possibility of differences in the timing of plagioclase crystallization as a potential driver behind differences in Eu enrichment in these samples. The timing of plagioclase crystallization should not significantly impact the mineral assemblage of the bulk granitic rock yet can influence the relative enrichment or depletion of trace elements such as Eu which are valuable indicators of these nuances in magmatic differentiation. We also interpret that these changes are relatively small when we consider the REE spider diagram shapes, which are similar between the two cobbles despite the Eu and Er anomalies. If mineral assemblages were significantly different, we would expect larger shifts in the REE spider diagrams shown in Figure S8. As we stated before, the driver of these differences involves the timing of crystallization of plagioclase, for which Fe and Mn are minor components. Therefore, the most important driver of the Fe and Mn content of the sediments is controlled by a varying amount of accessory minerals to the overall composition of the bedrock. With biotite being the primary aluminosilicate favored to chemically weather in these systems (Anderson et al., 2000), then the varying concentration of biotite in the bedrock, and thus, Fe and Mn, is important. Further, we highlight throughout the manuscript and in the response to reviewers below that published work suggests that underlying bedrock geochemistry is not typically the driver of glacial meltwater geochemistry (Aciego et al., 2015). Instead, the water-rock interaction time (which itself is governed by climate and glacial geomorphology) is the primary driver of subglacial water geochemistry and we discuss the details throughout the manuscript in section "Glacial setting drives downstream sediment geochemistry" (Lines 333-431).

- b. Even if the two source rocks are broadly similar in mineralogical composition aside from variations in plagioclase content, differences in the relative abundance of other mineral phases could still exert a significant influence on the mobilization of trace elements—particularly under differing chemical weathering regimes. In fact, the measured concentrations of Fe and Mn in the two cobble samples differ substantially (e.g., 1019 vs. 424 $\mu\text{mol g}^{-1}$ d.w.s. for total particulate Fe, and 33.91 vs. 18.16 $\mu\text{mol g}^{-1}$ d.w.s. for total particulate Mn), suggesting that source rock variability may still be a non-negligible factor.

We unfortunately have no way of truly knowing the bedrock composition beneath each of these glaciers beyond what has been previously mapped for this region, since no ice to bedrock cores have been sampled from the Harding Icefield. We acknowledge that our cobbles do show some compositional variability, and we also acknowledge that cobbles may not be truly representative of the bulk bedrock composition underlying each of the glaciers. Importantly, higher concentrations of Fe and Mn in the cobble from AG is not matched by a similarly high flux of Fe and Mn in the dissolved phase, nor in the total amount of particulate matter in meltwater. In fact, we see an opposite trend – greater concentrations of dFe and dMn as well as total pFe and pMn found in NWG

despite lower concentrations of these metals in the cobble— which suggests that other factors such as water-rock interaction time (subglacial weathering regime) rather than geology is the driving influence behind the observed trends in dissolved metal concentrations. In previously published studies, it has been documented that the elemental concentrations of interest in fjord surface waters (Fe, P, Si) is driven by water-rock interaction time, which can be influenced both by climate and geomorphology (Aciego et al., 2015) rather than the underlying bedrock composition. In the case of Aciego et al. (2015), outlet glaciers draining the Greenland Ice Sheet were investigated for dissolved elemental concentrations under the context of dramatically different subglacial bedrock geology and geochemistry and differences in water-rock interaction time. Despite differences in bedrock lithology ranging from Proterozoic supracrustal and intrusive complexes to Archean gneisses, the glacial meltwater micronutrient composition was primarily driven by water-rock interaction time rather than the subglacial geology (Aciego et al., 2015). This underscores the point that large (or small) differences in bedrock geology should not be the primary driver of trace metal compositions of the dissolved component of glacial meltwater, and that instead; water-rock interaction time which can be governed by glacial geometry, geomorphology, and other climate sensitive feedbacks.

- c. Most importantly, I would question the assertion that the REE patterns are largely similar between the two cobble samples. In my view, the differences are quite pronounced: the cobble from Aialik Glacier (AG) exhibits a middle rare earth element (MREE)-enriched pattern, whereas the cobble from Northwestern Glacier (NWG) shows a heavy rare earth element (HREE)-depleted pattern. These distinct signatures appear to be clearly reflected in the REE patterns of the sediments delivered from each respective source, suggesting that source rock composition may indeed influence sediment geochemistry more strongly than acknowledged in the manuscript.

We acknowledge these differences, however, while there are slight enrichments/depletions depending on the normalization procedure chosen, the overall shapes of the two cobbles and sediments are broadly similar (see new supplemental figures below). We note that a variable bedrock (e.g., cobbles) would have unique pattern shape and normalized concentrations that are much greater than 0.4 difference (Gabrielli et al., 2009, 2010). Further, studies demonstrate that despite large differences in bedrock composition, water-rock interaction time is the dominant control on the geochemical signal of mineral dissolution (Aciego et al., 2015). More detail on this point is covered in previous responses to reviewer comments above.

Figure S8. Rare Earth Element (REE) spider diagrams for detrital fjord sediments and cobbles from both fjords. Cerium-normalized concentrations are often used to demonstrate variability in source provenance, while Yb-normalized concentrations account for differences sediment concentration in driving the observed trends.

- k) While most of the typographical errors highlighted previously have been corrected, a number of minor errors still persist. More critically, there are lots of inconsistencies between the chemical data presented in Supplementary Table S2 and their interpretation in the Results section. For instance, the manuscript states:

“Based on total particulate Al, a relatively immobile element found in crustal aluminosilicates, we find that NWG produces more sediment than AG as average fjord surface concentrations are nearly twice as high (NWG: $\sim 16,000 \text{ nmol kg}^{-1}$, $n = 10$; AG: $\sim 7,900 \text{ nmol kg}^{-1}$, $n = 10$).”

However, based on calculations from the dataset provided, the average Al concentrations appear to be approximately $7,100 \text{ nmol kg}^{-1}$ for NWG and $3,960 \text{ nmol kg}^{-1}$ for AG. This discrepancy raises concerns about the reliability of other quantitative interpretations in the manuscript, and similar inconsistencies can be found elsewhere in the data.

We remain confident in our calculations of the data presented in Table S2, however, more details about how the data were treated are now included in the text. Specifically, when calculating averages, we necessarily sum the two size classes of particles to estimate an average for the total particulate metal within each fjord.

When we reference a total particulate metal concentration with no indication of the particle size, we have included the following text for clarity:

“Based on the summation of both particle size classes (<5 μm and >5 μm) to estimate total particulate Al...” (Lines 213-214)

“While Al, Ti, and P concentrations scaled with total particulates between the two fjords, summing both particle size classes (<5 μm and >5 μm)...” (Lines 223-224)

In summary, although the authors demonstrate a strong understanding of the system under study and have made efforts to address reviewer feedback, the manuscript continues to lack the clarity, internal consistency, and cohesion required to effectively communicate the significance of the findings. Given these persistent issues, the manuscript does not currently meet the journal’s standards for publication and is not sufficiently accessible to a broad scientific audience. I therefore recommend rejection.

Minor comments

#1. Lines 156-168: The authors consider AG to be in a stable tidewater configuration; however, its glacial flux is approximately five times greater than that of NWG, which is in a retreating configuration. This relationship is not immediately intuitive, and additional clarification would be helpful.

The glacial flux includes meltwater and solid ice calving, which occurs when the glacier terminates in the ocean. The result of a larger catchment and accumulation at higher altitudes compensates the current loss of ice mass at the glacier terminus. We have added this additional information to the main text.

When we reference a total particulate metal concentration with no indication of the particle size, we have included the following text for clarity:

“Based on the summation of both particle size classes (<5 μm and >5 μm) to estimate total particulate Al...” (Lines 213-214)

“While Al, Ti, and P concentrations scaled with total particulates between the two fjords, summing both particle size classes (<5 μm and >5 μm)...” (Lines 223-224)

#2. Lines 165-166: What is the scientific significance of the peak glacier velocity observed during your sampling period?

Peak velocities should correspond with greatest of sediments as faster glaciers enhances the stress on the bedrock (or till layer), leading to increased erosion and mobilization of sediments. This has been both empirically modeled and observed across surveys of alpine glacier settings. Therefore, the sediment plumes sampled should reflect peak seasonal generation of subglacial sediments to the coastal ocean. However, subglacial drainage, which occurs during peak meltwater production later in the summer, would be most important for the sustained delivery of subglacial sediments. The seasonality was not investigated in our study, so this clause has been removed.

#3. Lines 173-175: At stations 7, 8, and 9 in NWG, NO₃ and PO₄ concentrations generally appear higher on the western flanks of the fjords, which seems to contradict your interpretation. I guess this might be influenced by glacial flux from nearby land-terminating glaciers?

We will discuss each nutrient concentration, within each fjord, separately. The anomalously high concentrations of NO₃⁻ at stations 6 (AG) and 7 (NWG) on the western flanks may be explained by two separate sources. In AG, at the terminus, subglacial meltwater entrains subsurface water within a turbulent buoyant plume. Therefore, high NO₃⁻ at the surface here can be explained by a subsurface source within the sediment plume. Lowest salinities observed within AG fjord are found at station 6 (~26 ppt) supporting this hypothesis. However, we would also expect phosphate to follow a similar pattern, as both nutrients are enriched in subsurface waters. We do not observe a similarly high concentration at this station, possibly explained by the high amounts of oxides within this fjord, which are capable of scavenging phosphate from the water column due to the high surface area (Berner, 1973).

Within NWG, we cannot explain surface concentrations by the same mechanism. Instead, we hypothesize that the anomalously high concentrations of both nutrients at station 7 may be as the reviewer has pointed out. An additional source of both nutrients is derived from local land-terminating glaciers, for which Northwestern fjord contains two located on the western flank, which would input terrestrial-derived material to the fjord surface.

We have added text to the section “Dissolved trace-metals and macronutrients” to explain the elevated macronutrients at western flank stations. (Lines 162-173)

#4. Lines 190-191: This appears to contrast with your earlier statement: 'There was no observable trend in dFe concentrations with respect to distance from the glacier terminus or water column depth; (Lines 184-

185).

We note that the decreasing trend in dFe is less obvious in AG, whereas concentrations of dFe are strongly attenuated across a similar salinity range in other systems. In addition to NWG being a large source of dFe due to strong chemical weathering, the figure also shows that there might be more than one source of dFe to Aialik fjord, where the highest concentration was observed in the proximity of Pederson Lagoon on the western flank of the fjord (stations 7 and 8). This point source could reflect high concentrations of dFe coming from a nearby land-terminating glacier (Pedersen Glacier identified in Figure 1a). However, by station 10, concentrations have gradually been reduced (Figure below). By extrapolating the dFe trend in AG, one could greatly underestimate the zero-salinity endmember concentration since oxidative loss of dFe occurs immediately at the grounding line, where subglacial meltwater mixes with oxic seawater and rapidly converts dFe to authigenic particles (Forsch et al., 2021; Mikucki and Priscu, 2007).

Figure S5. Dissolved Fe (left) and Mn (right) versus sea surface salinity.

We have added a sentence that summarizes this trend with salinity/distance from the glacier terminus:

“Compared to dFe, the dMn concentrations do not attenuate as strongly with distance from the glacier terminus and across the salinity range (Figure S5), implying a more conservative behavior of dMn and as a possible tracer for recent glacial meltwater input of dFe to the ocean (Michael et al., 2023).” (Lines 199-202)

#5. Lines 205-207: Before this statement, I recommend clarifying the primary reason for the elevated Mn concentrations in the inner fjord. Meanwhile, given the lack of Mn data from the inner fjord during the fall, it is difficult to draw conclusions about the potential role of a time lag in coastal exchange. As you previously noted, glacial discharge peaks during the fall, so it may be more appropriate to suggest that increased meltwater input delivers more Mn, thereby enhancing its influence in more distal areas, such as the continental shelf. It is also plausible that Mn concentrations in the inner fjord are higher during the fall.

As can be seen in the Figure in response to the reviewer’s comment #4, the dMn is only slightly elevated at lower salinities, and the decay in concentration is distinctly different from that of dFe, presumably because of increased solubility in the surface ocean due to photoreduction of Mn-oxides, which cannot be ruled out as being present in these environments (Sherrell et al., 2018). The increased Mn in the inner fjord is likely due to the influence of glacial meltwaters, which have high dMn:dFe ratios, resuspended marine sediments, and entrainment of deep seawater overlying anoxic sediments, all of which can be a large source of Mn to benthic waters and contribute to continental shelf waters (Annett et al., 2017; Forsch et al., 2021), with glacial meltwaters as the largest contributor late in the summer (Michael et al., 2023).

We agree that without invoking a transport process, higher concentrations in the coastal ocean could

scale with glacial meltwater input and result in high concentrations of dMn on the continental shelf in the fall. However, we also note the timing of our sampling occurs when subglacial meltwater has the highest concentration of dissolved solutes as interstitial fluids in subglacial sediments are “flushed”. This can be summarized as a hysteresis (Whitfield and Schreier, 1981), where concentrations in meltwater are highest early in the melt season through increased meltwater-rock interaction time, but increases in meltwater throughout the season means that most Mn delivered to the fjord occurs late in the season. If we assume some dMn is also derived from enriched benthic seawater, and in the absence of wind-driven upwelling, then the concentrations in the surface would increase in AG due to the presence of a subglacial buoyant meltwater plume, which entrains subsurface fjord waters. This mechanism is not expected to occur in NWG.

We have removed our speculation on coastal exchange processes.

#6. Lines 216-217, 224, 225...: not consistent with dataset presented in Supplementary Table.

The station average total particulate metals are calculated from the sums of small and large particles (total particulate metal = total particulate metal <5 μm + total particulate metal >5 μm). We have added information about calculating total particulate metal concentrations for clarity.

#7. Lines 290-292: Intuitively, it seems that prolonged interaction between high-surface-area sediments and meltwater would most likely occur during a stable phase of the glacier. Could you provide supporting references for the statement?

We believe because eroded sediments are transported seasonally, there is little opportunity to store sediments during a stable phase and not possible to increase the water-rock interaction time.

However, if eroded sediments are stored in the subglacial environment at higher altitude from the equilibrium line (the position of maximum erosion) and the ability to transport these sediments is limited, then it is possible that there is prolonged contact time with water (chemical weathering). This is a matter of the dynamics of sediment storage versus sediment transport in these two configurations (stable and retreating). Fresh sediments eroded directly from bedrock are transported in a stable configuration, whereas when a glacier retreats, there is less bedrock erosion and more transport of “older” sediments that were stored for long periods in the subglacial environment as till layers (Delaney and Adhikari, 2020; Delaney and Anderson, 2022; Herman et al., 2021 and references therein).

We attempted to address all of these statements of nuance with the following text:

“As subglacial meltwater flushes through bedrock and glacial till surfaces, chemical weathering fluxes are a byproduct of water-rock interaction time (Aciego et al., 2015). Mechanical weathering primarily occurs during periods of glacial advance (Alley, 1991), and produces high surface area sediments which when exposed to water for prolonged periods of time (e.g., chemically weathered) will result in higher dissolved metal concentrations.” (Lines 291-295)

#8. Lines 315-316: Please provide a clearer interpretation and relevant supporting references

The leach protocol used in this study accesses biogenic cells, a labile component of elements that are solubilized at the heating temperature of 90°C (Berger et al., 2008). It is often assumed that much of the labile phosphorus concentration determined in marine suspended particles is of biogenic origin (Al-Hashem et al., 2022; Planquette et al., 2013). The labile fraction is likely to contain phytoplankton cells in fjord surface waters, in addition to PO₄ scavenged onto authigenic oxide minerals. Therefore, similarity in the concentrations of LpP and TpP of both fjords allow us to infer that there are similar abundances of phytoplankton cells.

We have added the following sentences to the paragraph to add a clearer interpretation and

supporting references:

“The labile fraction is likely to contain phytoplankton cells in fjord surface waters, in addition to PO₄ scavenged onto authigenic oxide minerals. Therefore, similarity in the concentrations of LpP and TpP of both fjords allow us to infer that there are similar abundances of phytoplankton cells.” (Lines 319-322)

#9. Line 331: Please provide a clearer interpretation and relevant supporting references

Please see discussion for major issue #1 and references listed therein.

#10. Line 367: I believe that Fe is generally considered a mobile element, in contrast to Al and Ti.

Thank you, we agree that relative to Al and Ti, Fe is more mobile. Sediments in NWG have lost Fe and Al to a similar degree since the ratio is quite invariable, meaning that Fe and Al are similarly chemically weathered, despite a small enrichment in labile Fe (Fig. 4a). Large amounts of Al-oxides are evidenced by the variable and high Al/Ti ratios in the adsorbed labile particulate fraction (Fig. 4b), which we interpret as greater chemical weathering despite most of the sediments overlapping with the NWG cobble Al/Ti signature.

#11. Line 416: fluctuations

Corrected.

#12. Figure 4: The authors measured elemental concentrations from two cobbles, which show substantial differences according to the Supplementary Table in my view. However, in Figure 4, the elemental ratios for the cobbles are presented as a single value, and it is unclear how this value was derived. It does not appear to be an average, nor does it correspond to data from either individual cobble. For example, the total Al/Ti and labile Al/Ti ratios for the cobbles are both shown as approximately 35, which is inconsistent with the values reported for cobble 1 (total Al/Ti = 16.85, labile Al/Ti = 33.07) and cobble 2 (total Al/Ti = 22.02, labile Al/Ti = 131.66), as well as their averages. A similar inconsistency exists for the Fe/Al (~0.3) and Mn/Fe (~0.03) ratios presented in the figure. Clarification on how these single values were calculated or derived is needed.

Thank you for bringing this to our attention. We acknowledge this mistake; it was incorrectly stated in the Figure 4 caption that the gray lines corresponded to average upper continental crust ratios from Taylor and McLennan (1995). We have adjusted the figure to now reflect the ratios of the two individual cobbles measured in this study. They are indicated as vertical blue lines in the new Figure 4, and the revised caption reflects this.

References Cited:

- Aarons, S. M., Dauphas, N., Greber, N. D., Roskosz, M., Bouchez, J., Carley, T., Liu, X.-M., Rudnick, R. L. and Gaillardet, J.: Titanium transport and isotopic fractionation in the Critical Zone, *Geochim. Cosmochim. Acta*, 352, 175–193, doi:<https://doi.org/10.1016/j.gca.2023.05.008>, 2023.
- Aciego, S. M., Stevenson, E. I. and Arendt, C. A.: Climate versus geological controls on glacial meltwater micronutrient production in southern Greenland, *Earth Planet. Sci. Lett.*, 424, 51–58, doi:<https://doi.org/10.1016/j.epsl.2015.05.017>, 2015.
- Al-Hashem, A. A., Beck, A. J., Krisch, S., Menzel Barraqueta, J.-L., Steffens, T. and Achterberg, E. P.: Particulate Trace Metal Sources, Cycling, and Distributions on the Southwest African Shelf, *Global Biogeochem. Cycles*, 36(11), e2022GB007453, doi:<https://doi.org/10.1029/2022GB007453>, 2022.
- Alley, R. B.: Sedimentary processes may cause fluctuations of tidewater glaciers, *Ann. Glaciol.*, 15, 119–124, doi:DOI: 10.3189/1991AoG15-1-115-121, 1991.
- Anderson, S. P., Drever, J. I., Frost, C. D. and Holden, P.: Chemical weathering in the foreland of a retreating glacier, *Geochim. Cosmochim. Acta*, 64(7), 1173–1189, doi:[https://doi.org/10.1016/S0016-7037\(99\)00358-0](https://doi.org/10.1016/S0016-7037(99)00358-0), 2000.
- Annett, A. L., Fitzsimmons, J. N., Séguret, M. J. M., Lagerström, M., Meredith, M. P., Schofield, O. and Sherrell, R. M.: Controls on dissolved and particulate iron distributions in surface waters of the Western

Antarctic Peninsula shelf, *Mar. Chem.*, doi:10.1016/j.marchem.2017.06.004, 2017.

Berger, C. J. M., Lippiatt, S. M., Lawrence, M. G. and Bruland, K. W.: Application of a chemical leach technique for estimating labile particulate aluminum, iron, and manganese in the Columbia River plume and coastal waters off Oregon and Washington, *J. Geophys. Res. Ocean.*, 113(C2), doi:https://doi.org/10.1029/2007JC004703, 2008.

Berner, R. A.: Phosphate removal from sea water by adsorption on volcanogenic ferric oxides, *Earth Planet. Sci. Lett.*, 18(1), 77–86, doi:https://doi.org/10.1016/0012-821X(73)90037-X, 1973.

Bouchez, J., Lupker, M., Gaillardet, J., France-Lanord, C. and Maurice, L.: How important is it to integrate riverine suspended sediment chemical composition with depth? Clues from Amazon River depth-profiles, *Geochim. Cosmochim. Acta*, 75(22), 6955–6970, doi:https://doi.org/10.1016/j.gca.2011.08.038, 2011.

Delaney, I. and Adhikari, S.: Increased Subglacial Sediment Discharge in a Warming Climate: Consideration of Ice Dynamics, Glacial Erosion, and Fluvial Sediment Transport, *Geophys. Res. Lett.*, 47(7), e2019GL085672, doi:https://doi.org/10.1029/2019GL085672, 2020.

Delaney, I. and Anderson, L. S.: Debris Cover Limits Subglacial Erosion and Promotes Till Accumulation, *Geophys. Res. Lett.*, 49(16), e2022GL099049, doi:https://doi.org/10.1029/2022GL099049, 2022.

Dumoulin, J. A.: Sandstone composition of the Valdez and Orca Groups, Prince William Sound, Alaska: U.S. Geological Survey Bulletin 1774., 1987.

Eidam, E. F., Nittrouer, C. A., Lundesgaard, Homolka, K. K. and Smith, C. R.: Variability of Sediment Accumulation Rates in an Antarctic Fjord, *Geophys. Res. Lett.*, doi:10.1029/2019GL084499, 2019.

Föllmi, K. B., Arn, K., Hosein, R., Adatte, T. and Steinmann, P.: Biogeochemical weathering in sedimentary chronosequences of the Rhône and Oberaar Glaciers (Swiss Alps): Rates and mechanisms of biotite weathering, *Geoderma*, 151(3), 270–281, doi:https://doi.org/10.1016/j.geoderma.2009.04.012, 2009.

Forsch, K. O., Hahn-Woernle, L., Sherrell, R. M., Roccanova, V. J., Bu, K., Burdige, D., Vernet, M. and Barbeau, K. A.: Seasonal dispersal of fjord meltwaters as an important source of iron and manganese to coastal Antarctic phytoplankton, *Biogeosciences*, 18(23), 6349–6375, doi:10.5194/bg-18-6349-2021, 2021.

Gabrielli, P., Planchon, F., Barbante, C., Boutron, C. F., Petit, J. R., Bulat, S., Hong, S., Cozzi, G. and Cescon, P.: Ultra-low rare earth element content in accreted ice from sub-glacial Lake Vostok, Antarctica, *Geochim. Cosmochim. Acta*, 73(20), 5959–5974, doi:https://doi.org/10.1016/j.gca.2009.05.050, 2009.

Gabrielli, P., Wegner, A., Petit, J. R., Delmonte, B., De Deckker, P., Gaspari, V., Fischer, H., Ruth, U., Kriews, M., Boutron, C., Cescon, P. and Barbante, C.: A major glacial-interglacial change in aeolian dust composition inferred from Rare Earth Elements in Antarctic ice, *Quat. Sci. Rev.*, 29(1), 265–273, doi:https://doi.org/10.1016/j.quascirev.2009.09.002, 2010.

Herman, F., De Doncker, F., Delaney, I., Prasicek, G. and Koppes, M.: The impact of glaciers on mountain erosion, *Nat. Rev. Earth Environ.*, 2(6), 422–435, doi:10.1038/s43017-021-00165-9, 2021.

Kryc, K. A., Murray, R. W. and Murray, D. W.: Al-to-oxide and Ti-to-organic linkages in biogenic sediment: relationships to paleo-export production and bulk Al/Ti, *Earth Planet. Sci. Lett.*, 211(1), 125–141, doi:https://doi.org/10.1016/S0012-821X(03)00136-5, 2003.

Michael, S. M., Crusius, J., Schroth, A. W., Campbell, R. and Resing, J. A.: Glacial meltwater and sediment resuspension can be important sources of dissolved and total dissolvable aluminum and manganese to coastal ocean surface waters, *Limnol. Oceanogr.*, 68(6), 1201–1215, doi:https://doi.org/10.1002/lno.12339, 2023.

Mikucki, J. A. and Priscu, J. C.: Bacterial diversity associated with blood falls, a subglacial outflow from the Taylor Glacier, Antarctica, *Appl. Environ. Microbiol.*, doi:10.1128/AEM.01396-06, 2007.

Naney, M. T. and Swanson, S. E.: The effect of Fe and Mg on crystallization in granitic systems, *Am. Mineral.*, 65(7–8), 639–653, 1980.

Nesbitt, H. W.: Mobility and fractionation of rare earth elements during weathering of a granodiorite, *Nature*, 279(5710), 206–210, doi:10.1038/279206a0, 1979.

Planquette, H., Sherrell, R. M., Stammerjohn, S. and Field, M. P.: Particulate iron delivery to the water column of the Amundsen Sea, Antarctica, *Mar. Chem.*, doi:10.1016/j.marchem.2013.04.006, 2013.

Pryer, H. V., Hawkings, J. R., Wadham, J. L., Robinson, L. F., Hendry, K. R., Hatton, J. E., Kellerman, A. M., Bertrand, S., Gill-Olivas, B., Marshall, M. G., Brooker, R. A., Daneri, G. and Häussermann, V.: The Influence of Glacial Cover on Riverine Silicon and Iron Exports in Chilean Patagonia, *Global Biogeochem. Cycles*, 34(12), e2020GB006611, doi:https://doi.org/10.1029/2020GB006611, 2020.

Schroth, A. W., Crusius, J., Chever, F., Bostick, B. C. and Rouxel, O. J.: Glacial influence on the geochemistry of riverine iron fluxes to the Gulf of Alaska and effects of deglaciation, *Geophys. Res. Lett.*, 38(16), doi:10.1029/2011GL048367, 2011.

Sherrell, R. M., Annett, A. L., Fitzsimmons, J. N., Roccanova, V. J. and Meredith, M. P.: A “shallow bathtub ring” of local sedimentary iron input maintains the Palmer Deep biological hotspot on the West Antarctic Peninsula shelf, *Philos. Trans. R. Soc. A Math. Phys. Eng. Sci.*, doi:10.1098/rsta.2017.0171, 2018.

Syvitski, J. P. M.: On the deposition of sediment within glacier-influenced fjords: Oceanographic controls, *Mar. Geol.*, 85(2), 301–329, doi:https://doi.org/10.1016/0025-3227(89)90158-8, 1989.

Tysdal, R.G.; Case, J. E.: Geologic map of the Seward and Blying Sound quadrangles, Alaska: U.S. Geological Survey Miscellaneous Investigations Series Map I-1150, scale 1:250,000., 1979.

Wei, G., Li, X.-H., Liu, Y., Shao, L. and Liang, X.: Geochemical record of chemical weathering and monsoon climate change since the early Miocene in the South China Sea, *Paleoceanography*, 21(4), doi:<https://doi.org/10.1029/2006PA001300>, 2006.

Whitfield, P. H. and Schreier, H.: Hysteresis in relationships between discharge and water chemistry in the Fraser River basin, British Columbia, *Limnol. Oceanogr.*, 26(6), 1179–1182, doi:<https://doi.org/10.4319/lo.1981.26.6.1179>, 1981.

Zhao, K. X., Stewart, A. L., McWilliams, J. C., Fenty, I. G. and Rignot, E. J.: Standing Eddies in Glacial Fjords and Their Role in Fjord Circulation and Melt, *J. Phys. Oceanogr.*, 53(3), 821–840, doi:<https://doi.org/10.1175/JPO-D-22-0085.1>, 2023.